# Sarcolemmal and mitochondrial membrane potentials measured *ex vivo* and *in vivo* in the heart by pharmacokinetic modelling of [99mTc]sestamibi

Edward C. T. Waters[1], Friedrich Baark[1], Matthew R. Orton[2,3], Michael J. Shattock[4,5] (ID),
Richard Southworth[1,5] (ID) and Thomas R. Eykyn[1,5] (ID)

[1]*School of Biomedical Engineering and Imaging Sciences, King's College London, The Rayne Institute, St Thomas' Hospital, London, UK*
[2]*Department of Radiology, MRI Unit, The Royal Marsden NHS Foundation Trust, London, UK*
[3]*Division of Radiotherapy and Imaging, The Institute of Cancer Research, London, UK*
[4]*School of Cardiovascular and Metabolic Medicine and Sciences, King's College London, London, UK*
[5]*King's College London BHF Centre of Research Excellence, King's College London, London, UK*

Handling Editors: Natalia Trayanova & Brian Delisle

The peer review history is available in the Supporting information section of this article
(https://doi.org/10.1113/JP290295#support-information-section).

**Abstract figure legend** The pharmacokinetics of lipophilic cationic radiotracers such as [99mTc]sestamibi and [99mTc]tetrofosmin are dependent on the electrical potentials that determine their distribution across the sarcolemmal and mitochondrial membranes. Fitting their temporal kinetics to a mathematical model that is reparametrized using the Nernst equation allows an estimation of these voltages *ex vivo* in the Langendorff perfused heart under baseline conditions, under hyperkalaemic depolarization and mitochondrial uncoupling with carbonyl cyanide 3-chlorophenylhydrazone. Using planar scintigraphy, the method can be extended to give an independent estimation of these fundamental physiological parameters *in vivo*. Created in BioRender.

**Edward Waters** completed his PhD in the Department of Imaging Chemistry and Biology in the School of Biomedical Engineering and Imaging Sciences at King's College London. He went on to pursue a Postdoctoral Fellowship in Cardiac Molecular Imaging at King's College London. His research focuses on developing targeted probes to visualize cardiovascular pathology and novel imaging techniques in cardio-oncology. A career highlight was being awarded a grant from the Wellcome EPSRC Centre for Medical Engineering (CME) Postdoctoral Support Scheme grant to develop novel positron emission tomography radiopharmaeuticals for non-invasive *in vivo* REDOX imaging, including translating these molecular imaging tools into the clinical setting to better understand cardiovascular diseases.

The Journal of Physiology

**Abstract** Transmembrane electrical potentials across the sarcolemmal ($E_m$) and mitochondrial ($\Delta\Psi_m$) membranes are central to cellular excitability, metabolism and viability. However, their direct and quantitative measurement *in vivo* remains challenging. We established a quantitative kinetic modelling framework to estimate $E_m$ and $\Delta\Psi_m$ independently from dynamic radiotracer data in the heart using the Nernst equation applied to the kinetics of the lipophilic cationic tracers [99mTc]sestamibi and [99mTc]tetrofosmin. Parameters were estimated from high-temporal-resolution time–activity curves using non-linear least squares and Markov chain Monte Carlo (MCMC) fitting. Experiments were performed in isolated Langendorff-perfused rat hearts under baseline, hyperkalaemic depolarization and mitochondrial uncoupling with carbonylcyanide-3-chlorophenylhydrazone (CCCP) and *in vivo* using planar scintigraphy. In perfused hearts, baseline potentials were $E_m = -65 \pm 7$ mV and $\Delta\Psi_m = -109 \pm 9$ mV (mean $\pm$ SD, $n = 4$). Increasing [K$^+$] caused dose-dependent depolarization of $E_m$ in agreement with Goldman–Hodgkin–Katz predictions, whereas $\Delta\Psi_m$ remained stable. CCCP selectively depolarized $\Delta\Psi_m$ to $-66 \pm 8$ mV (300 nM) and $-6 \pm 2$ mV (600 nM) with minimal effect on $E_m$. *In vivo*, potentials were $E_m = -61 \pm 8$ mV and $\Delta\Psi_m = -151 \pm 13$ mV ($n = 4$), consistent with physiological values. This modelling approach enables the first non-invasive, independent quantitative estimation of sarcolemmal and mitochondrial membrane potentials *in vivo*. It overcomes limitations of optical probes and, with high-sensitivity single-photon emission computed tomography and positron emission tomography (PET) systems (including total body PET), offers new opportunities to assess bioenergetic dysfunction in cardiovascular disease and beyond.

(Received 9 October 2025; accepted after revision 9 February 2026; first published online 16 March 2026)

**Corresponding authors** R. Southworth and T. R. Eykyn: School of Biomedical Engineering and Imaging Sciences, King's College London, The Rayne Institute, St Thomas' Hospital, London SE1 7EH, UK. Email: thomas.eykyn@kcl.ac.uk; richard.southworth@kcl.ac.uk

## Key points

- Pharmacokinetic modelling of [99mTc]sestamibi and [99mTc]tetrofosmin allowed independent estimation of sarcolemmal ($E_m$) and mitochondrial ($\Delta\Psi_m$) membrane potentials *ex vivo* in the Langendorff perfused rat heart and *in vivo* in the rat heart.
- The method gave independent measures of membrane potentials *ex vivo* when depolarized with hyperkalaemic buffers or mitochondrial uncoupling.
- *In vivo* measurements of membrane potentials agreed with literature values, whereas $\Delta\Psi_m$ was found to be less polarized *ex vivo* in the perfused heart.
- The method uses clinically available single-photon emission computed tomography imaging agents that could be employed to measure these parameters in humans.

## Introduction

First investigated by Hodgkin, Katz and Huxley (Hodgkin & Katz, 1949; Hodgkin et al., 1952) and Mitchell (Mitchell, 1961), transmembrane electrical potentials across the sarcolemmal ($E_m$) and inner mitochondrial membranes ($\Delta\Psi_m$) are essential features of nearly all mammalian cells. They play a critical role in normal cell function and are perturbed in response to many disease processes. In the healthy heart sarcolemmal membrane potential plays a key role in normal electrical and mechanical function and contributes to systolic and diastolic dysfunction in ischaemia, heart failure and arrhythmias. In cancer cells the transmembrane potential is known to be critical in cell cycle progression (Yang & Brackenbury, 2013) and, via changes in ion flux, may contribute to altered metabolism (Michaels et al., 2024). On the other hand, mitochondrial membrane potentials $\Delta\Psi_m$ are normally maintained in the range $-135$ to $-165$ mV (Kowaltowski & Abdulkader, 2024) but can be depolarized or hyperpolarized under pathophysiological conditions. This potential arises from ion gradients and the electron transport chain, which pumps H$^+$ from the mitochondrial matrix into the inter membrane space, generating both an electrochemical and pH gradient (matrix $\sim 0.5$pH units more alkaline (Perry et al., 2011)). Together, these components form

the proton motive force ($\Delta p$) that drives ATP synthesis via the $F_0F_1$-ATPase (Mitchell, 1961). So critical is $\Delta \Psi_m$ to cell energetics and viability, hyperpolarizing the mitochondrial membrane potential using a novel optogenetic approach can significantly increase lifespan in *Caenorhabditis elegans* (Berry et al., 2023).

Because of their fundamental importance in health and disease, dynamically quantifying changes in these membrane potentials *in vivo* is of great experimental and clinical interest, but non-invasive techniques for doing so *in vivo* are currently very limited. Measurement of the plasma membrane potential ($E_m$) is mostly restricted to experiments in isolated cells or tissues using voltage-clamping, sharp micro-electrodes or voltage-sensitive dyes (Snabaitis et al., 1997). The mitochondrial membrane potential $\Delta \Psi_m$ is commonly measured in cultured cells *in vitro* using lipophilic cationic probes (Kamo et al., 1979; Rottenberg, 1984) such as tetramethylrhodamine ethyl ester (TMRE), tetraphenylphosphonium (TPP), triphenylmethylphosphonium (TPMP) and tetraphenylarsonium (TPA) (Kowaltowski & Abdulkader, 2024), for which the subcellular distribution at equilibrium can be interpreted using the Nernst equation. However, their use *in vivo* is confounded by the fact that these probes must be used at relatively high concentrations to elicit a measurable signal, often to the point that they concentrate within mitochondria to such an extent that they pi-stack and quench their own signal at high concentrations (Kowaltowski & Abdulkader, 2024). Experiments are therefore usually either performed in 'quenching' or 'non-quenching' mode as fluorophores either accumulate or leach out of mitochondria according to their membrane potentials. The measurements that they provide are hard to calibrate, prone to artefacts and difficult to interpret *in vitro*, whereas their usefulness in intact perfused tissues (or indeed *in vivo*) is very limited because of fluorophore toxicity, poor tissue penetration by light and a lack of whole-body scanners, etc.

Non-invasive nuclear 'molecular' imaging approaches such as single-photon emission computed tomography (SPECT) and positron emission tomography (PET) are therefore increasingly being explored to address this challenge. These exquisitely sensitive techniques allow the non-invasive quantification and imaging of radiotracer pharmacokinetics at subpharmacological (typically sub-nanomolar) concentrations without perturbing the system under investigation. The SPECT tracers [99mTc]sestamibi and [99mTc]tetrofosmin are widely used clinical myo-cardial perfusion imaging agents (Berman et al., 1994). They are lipophilic cations and therefore their cellular uptake depends on both sarcolemmal and mitochondrial potentials (Kawamoto et al., 2015). Sestamibi uptake is sensitive to mitochondrial uncoupling with carbonyl cyanide *m*-chlorophenylhydrazone (CCCP), with

comparable sensitivity to TMRE (Kawamoto et al., 2015) and to doxorubicin-induced cardiotoxicity, independent of perfusion effects (Safee et al., 2019). Crucially for this application, they are not confounded by the concentration-dependent limitations with respect to quenching and toxicity which hampers fluorescence techniques and allow whole body radio-tracer imaging irrespective of tissue depth. Comparable PET approaches using other lipophilic cations [11C]TPMP (Fukuda et al., 1986), [18F]TPP (Gurm et al., 2012) or [4-18F]fluorobenzyl-triphenylphosphonium (18F-BnTP) (Momcilovic et al., 2019) have also been explored, in combination with contrast-enhanced magnetic resonance imaging (MRI) with gadolinium to estimate volume fractions (Alpert et al., 2018; Pelletier-Galarneau et al., 2021) but, to date, these techniques do not resolve sarcolemmal and mitochondrial contributions, and have been limited to measuring total tissue membrane potential $E_t = E_m + \Delta \Psi_m$.

The present study aimed to explore a different approach involving kinetic modelling of dynamic data using the Nernst equation to numerically fit time-activity curves to obtain independent *in vivo* estimates of sarcolemmal and mitochondrial membrane potentials. The model describes [99mTc]sestamibi and [99mTc]tetrofosmin (Fig. 1) uptake kinetics in isolated Langendorff-perfused hearts under baseline conditions, and when either (or both) membranes are depolarized pharmacologically. Parameters were estimated using least squares fitting combined with a Markov chain Monte Carlo (MCMC) approach, enabling systematic random sampling of the parameter space (Kuchel et al., 2011). Applying this framework to high-temporal-resolution single-photon emission planar scintigraphy allowed us to address a long-standing challenge in molecular cardiology and mitochondrial biology: direct, non-invasive measurement of electrical potentials across distinct membranes in intact organs. The method has immediate translational potential in humans and could extend the role of these radiotracers beyond perfusion assessment to provide quantitative insights into mitochondrial dysfunction and

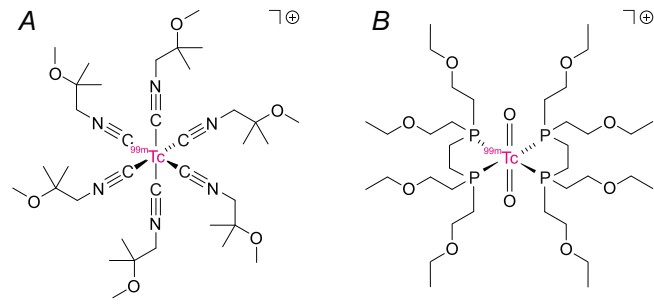

**Figure 1. Molecular structures of the radiotracers**
*A*, [99mTc]sestamibi. *B*, [99mTc]tetrofosmin.

cell membrane potentials as hallmarks of cardiovascular disease, metabolic syndrome, neurodegeneration and cancer.

To our knowledge, this is the first time that dynamic pharmacokinetic modelling of lipophilic cationic radiotracers has been used to independently and quantitatively resolve sarcolemmal and mitochondrial membrane potentials *in vivo*. The increasing availability of high-sensitivity dynamic SPECT and total-body PET systems will allow whole-organ and potentially whole-body mapping of membrane potentials with high temporal resolution with a fraction of the radiation dose that has previously been required (Badawi et al., 2019; Cherry et al., 2018; Cook et al., 2025; Zhang et al., 2020). This opens new opportunities for early detection of metabolic dysfunction, treatment monitoring, personalized risk stratification, bridging basic mitochondrial physiology with precision medicine.

## Theory

### Goldman–Hodgkin–Katz (GHK) and Nernst equations

The sarcolemmal membrane potential ($E_m$ in V) is given by the ratio of the equilibrium concentrations of the major permeant ions ($K^+$, $Na^+$, $Cl^-$) and their relative permeabilities, as described by the GHK equation,

$$E_m = \frac{RT}{F} \ln\left(\frac{P_K[K^+]_{out} + P_{Na}[Na^+]_{out} + P_{Cl}[Cl^-]_{in}}{P_K[K^+]_{in} + P_{Na}[Na^+]_{in} + P_{Cl}[Cl^-]_{out}}\right), \quad (1)$$

where the universal gas constant is $R = 8.314 \text{ J mol}^{-1} \text{ K}^{-1}$, the Faraday constant $F = 96{,}485 \text{ C mol}^{-1}$ (kJ V$^{-1}$ mol$^{-1}$) and the temperature $T = 310$ K. Typical concentrations in cardiac tissue are (mM) $[K^+]_{out} = 4.9$, $[K^+]_{in} = 115$, $[Na^+]_{out} = 140$, $[Na^+]_{in} = 15$, $[Cl^-]_{out} = 115$ and $[Cl^-]_{in} = 15$, with relative permeabilities at rest of $P_K = 1$, $P_{Na} = 0.01$ and $P_{Cl} = 0.1$ yielding a calculated resting membrane potential of $E_m = -75$ mV. The relative permeabilities are dependent on ion channel expression as well as the opening or closing of voltage gated ion channels (Veech et al., 2002). For example, the plasma membrane is relatively impermeable to $Na^+$ at resting potentials and significantly increases during the action potential leading to depolarization of the membrane potential. Using eqn (1), the sarcolemmal membrane potential can be calculated over a range of $[K^+]_{out}$ values for different experimental Krebs–Henseleit buffer (KHB) solutions and assuming the other parameter values are constant.

### Three-compartment model describing [$^{99m}$Tc]sestamibi pharmacokinetics

The compartmental model shown in Fig. 2 describes the delivery of an arterial bolus input $U(t)$ to give a

plasma concentration $c_p(t)$ which can leave the tissue via the venous outflow or diffuse across the sarcolemmal membrane with a rate constant $k_1$. The concentration of molecules in the cytosol $c_c(t)$ can diffuse back out of the cell with a rate constant $k_{-1}$, or they can diffuse into or out of the mitochondria with rate constants $k_2$ and $k_{-2}$, respectively, to give a mitochondrial concentration $c_m(t)$. Note that standard nomenclature in radiotracer compartmental modelling literature uses $k_2$ instead of $k_{-1}$, and $k_3$ and $k_4$ instead of $k_2$ and $k_{-2}$. We retain the use of standard chemical nomenclature here since $k_1$ and $k_{-1}$ describe the equilibrium constant across the plasma membrane $K_1$, whereas $k_2$ and $k_{-2}$ describe the equilibrium constant across the mitochondrial membrane $K_2$ (see below).

Assuming ionic solutes are present at tracer concentration, and the membrane potential is not perturbed by their redistribution, then the ratio of the chemical activities across the membrane at steady state depends on the membrane potential described be the Nernst equation:

$$E = \frac{RT}{zF} \ln\left(\frac{a_{out}}{a_{in}}\right), \quad (2)$$

where $z$ is the ionic charge, in the case of [$^{99m}$Tc]sestamibi $z = +1$.

Note that the Nernst equation is expressed in terms of the reaction quotient of chemical activities $a_{out}/a_{in}$

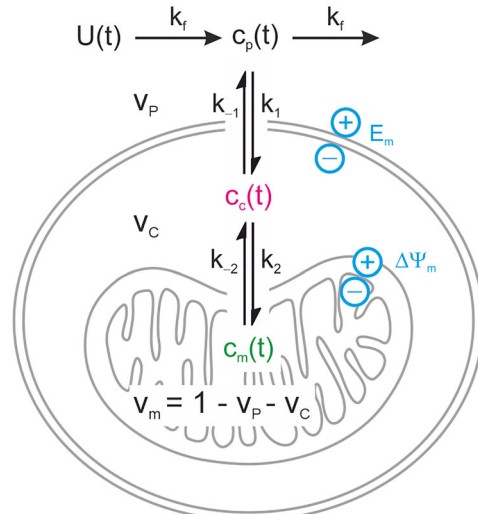

**Figure 2. Model framework**
Three compartment model describing radiotracer passage through the heart: plasma ($c_p(t)$, black), cytosol ($c_c(t)$, magenta) and mitochondria ($c_m(t)$, green). The bolus input function $U(t)$ enters the plasma compartment via flow (rate constant $k_f$). Exchange across the sarcolemmal membrane is described by $k_1$ and $k_{-1}$, and across the mitochondrial membrane by $k_2$ and $k_{-2}$. Sarcolemmal and mitochondrial membrane potentials are represented by $E_m$ and $\Delta\Psi_m$.

which accounts for non-ideal behaviour. However, noting that the concentrations of radioactive species are typically in the nM range, see Methods, then the chemical activities are equal to chemical concentrations in very dilute solutions, and the reaction quotient is equal to the equilibrium constant $K$. Non-specific binding of the tracer inside the cell could lead to a deviation from ideal behaviour. P-glycoprotein (P-gp) expression can also lead to efflux of the tracer in some disease models which may also lead to non-ideal behaviour, however basal expression of P-gp in the myocardium is very low (Couture et al., 2006).

From standard chemical kinetics, the equilibrium constant is given by the ratio of the rate constants $K_1 = k_1/k_{-1}$ and $K_2 = k_2/k_{-2}$ even for a system not at steady state. The reverse rate constants $k_{-1}$ and $k_{-2}$ can therefore be expressed as a function of the forwards rate constants and the corresponding membrane potential by rearranging the Nernst equation.

$$k_{-1} = k_1 \, \exp\left(\frac{FE}{RT}\right). \qquad (3)$$

The system of differential equations describing the model in Fig. 2 can then be expressed as a function of concentrations and membrane potentials as:

$$\frac{dc_p(t)}{dt} = k_f \, U(t) - \left(k_1 + k_f + k'\right) c_p(t)$$
$$+ k_1 \exp\left(\frac{FE_m}{RT}\right) c_c(t), \qquad (4)$$

$$\frac{dc_t(t)}{dt} = k_1 \, c_p(t) - \left(k_1 \exp\left(\frac{FE_m}{RT}\right) + k_2 + k'\right) c_c(t)$$
$$+ k_2 \exp\left(\frac{F\Delta\Psi_m}{RT}\right) c_m(t), \qquad (5)$$

$$\frac{dc_m(t)}{dt} = k_2 \, c_c(t) - \left(k_2 \exp\left(\frac{F\Delta\Psi_m}{RT}\right) + k'\right) c_m(t), \qquad (6)$$

where $E_m$ and $\Delta\Psi_m$ are the sarcolemmal and mitochondrial membrane potentials, respectively, which are defined as negative voltages (inside the membrane) in the above equations. The first order rate constant $k' = 3.2059 \times 10^{-5}$ s$^{-1}$ describes the radioactive decay rate constant of $^{99m}$Tc ($k' = \ln 2 \, /\lambda$ where $\lambda = 6.006$ h is the half-life of $^{99m}$Tc). Radioactivity is measured in counts per second (cps) and is not a direct measure of concentration. However, activity per unit compartment volume is proportional to concentration (mol per unit volume). Providing the kinetic model is first order, as is the case here, then the rate constants $k$ (units s$^{-1}$) are independent of concentrations.

The bolus input function is represented by a modified gamma-variate function given by:

$$U(t) = A_0 (t - t_0)^\alpha \exp\left(-\beta (t - t_0)\right), \qquad (7)$$

where $t_0$ is the initial arrival time of the bolus. $A_0$ is a normalization factor, and $\alpha$ and $\beta$ characterize the rise and fall of the concentration of the input bolus, respectively (i.e. the width or dispersion of the bolus). The measured signal (radioactivity) is the volume-weighted average of the three compartments, given by:

$$S(t) = V_p c_p(t) + V_c c_c(t) + \left(1 - V_p - V_c\right) c_m(t) + B, \qquad (8)$$

where $B$ is a fitting parameter that allows for non-zero baseline offset due to background radiation. Volume fractions of the plasma, cytosolic and mitochondrial compartments are $V_p$, $V_c$ and $(1 - V_p - V_c)$, respectively.

For flux of a solute into the tissue the rate constant $k_1$ (typically referred to as $K_1$, not to be confused with the equilibrium constant) depends on blood flow and extraction fraction, with expressions derived by Renkin and Cone (Crone, 1963; Renkin, 1959). If the membrane permeability is low and/or the surface area of the capillary membrane is small compared to flow, then the rate constant is independent of flow and becomes dependent on membrane transport. Membrane permeability of [$^{99m}$Tc]sestamibi is lower than typical flow tracers and is independent of flow at high perfusion rates, only becoming flow dependent when perfusion decreases; for example, during ischaemia (Kontos et al., 1997). Normal coronary arterial perfusion in the human heart is 2.4 mL g$^{-1}$ min$^{-1}$ (Maddahi & Packard, 2014) with slightly higher values in the rat heart *in vivo* (Hershgold et al., 1959). The rate of arterial perfusion in the Langendorff perfused rat heart is much higher at 14 mL g$^{-1}$ min$^{-1}$. When perfusion is not rate limiting, then the rate constant $k_1$ is dependent on diffusion mediated transport, which depends on the membrane potential.

## Methods

### Ethical approval

All experimental procedures were approved by King's College London's local Animal Care and Ethics Committee and carried out in accordance with Home Office regulations as detailed in the Guidance on the Operation of Animals (Scientific Procedures) Act 1986 under Home Office licence number PP8261525. The study conformed to the ethical principles of *The Journal of Physiology*.

## Reagents and gas mixtures

All reagents were purchased from Sigma-Aldrich (St Louis, MO, USA) unless otherwise stated. All gas mixtures were purchased from BOC Industrial Gases (Woking, UK).

## Heart excision and perfusion

Adult male Wistar rats (275–325 g) were purchased from Charles River Laboratories (Margate, UK). Animals were maintained under a 12:12 h light/dark photocycle at 22 ± 2°C with access to standard chow diet and water *ad libitum*. Animals were terminally anaesthetized by I.P. injection of sodium pentobarbital (200 mg kg$^{-1}$) and sodium heparin (200 IU kg$^{-1}$). Surgical level of anaesthesia was confirmed by absence of corneal and pedal reflexes and hearts were then excised and immediately arrested in ice-cold KHB solutions composed of (in mM): 118 NaCl, 5.9 KCl, 1.16 MgSO$_4$, 25 NaHCO$_3$, 0.48 NaEDTA, 11.1 glucose and 2.2 CaCl$_2$. Hearts were then cannulated via the aorta and secured using a 3–0 suture (Ethicon, Raritan, NJ, USA). The pulmonary artery was incised to allow coronary effluent drainage, and hearts were perfused with KHB equilibrated with 95% O$_2$/5% CO$_2$ at 37°C. Perfusion was maintained at a constant flow of 14 mL min$^{-1}$. Contractile function was monitored using an intraventricular balloon (IVB) adjusted to an initial end-diastolic pressure of 4–10 mmHg. Perfusion pressure and cardiac function were continuously recorded with two pressure transducers connected to a PowerLab system (AD Instruments Ltd, Bella Vista, NSW, Australia). After stabilization, perfusion was switched to one of the following treatments: vehicle control (0.02% v/v ethanol), CCCP (300–600 nM), hyperkalaemic KHB (4.9–25 mM KCl) or their combination. Ten minutes after treatment, a single bolus of [$^{99m}$Tc]sestamibi or [$^{99m}$Tc]tetrofosmin (5 MBq in 50 µL at ∼5 nM) was injected via the perfusion line.

## The triple $\gamma$-detector system

Radiotracer pharmacokinetics were monitored using a triple $\gamma$-detector system, as previously described (Handley et al., 2014; Medina et al., 2015). The setup comprised three orthogonal lead-collimated Na/I $\gamma$-detectors arrayed around the Langendorff isolated heart perfusion apparatus: (i) on the arterial line 3 cm downstream of the injection port and 15 cm upstream of the heart cannula; (ii) directly opposite the heart; and (iii) over the venous outflow line. Each detector was connected to a modified GinaSTAR$^{TM}$ system running Gina$^{TM}$ software (Raytest Ltd, Chessington, UK) for real-time signal collection. Cardiac tracer uptake was monitored

with 200 ms temporal resolution over a 30 min recording period.

## *In vivo* planar scintigraphy

Animals were anaesthetized in an induction chamber with isoflurane (4% v/v at a flow rate of 0.5–1 L min$^{-1}$) and maintained under anaesthesia via a nose cone (2% v/v isoflurane at a flow rate of 0.5–1 L min$^{-1}$). Rats were secured with tape and positioned supine on a heating pad (MouseMonitor S; Indus Instruments Webster, TX, USA) to maintain body temperature at 37°C and monitored using a rectal probe (MLT1403; AD Instruments Ltd). Nuclear imaging was performed using a Nanoscan SPECT/CT scanner (Mediso, Budapest, Hungary). Following transfer to the scanner, animals received an I.V. injection of radiotracer (100 MBq in 1 mL of sterile saline) via the femoral vein. Dynamic scanning was initiated ∼60 s before injection to capture the complete time course, including the bolus phase. Planar imaging consisted of 2.5 h of dynamic acquisition (1 s frames for the first 10 min, 5 s frames until 30 min, and 10 s frames for the final 2 h) followed by a 10 min static scan. At the end of scanning animals were culled by overdose of anaesthetic. Images were processed and analyzed using VivoQuant$^{TM}$ software (Invicro, Needham, MA, USA). A region of interest (ROI) was manually placed around the heart and applied consistently across all images and timepoints.

## Least squares fitting

Time–activity curves sampled at times $t_1$, $t_2$, …, $t_n$ were normalized to peak activity and simultaneously fit to the system of differential equations (eqns 4–8) using the *lsqnonlin* non-linear least squares algorithm with the ODE solver *ode23s* in MATLAB (MathWorks Inc., Natick, MA, USA). Twelve parameters were varied freely within defined boundary conditions: $k_f$, $k_1$, $E_m$, $k_2$, $\Delta\Psi_m$, $t_0$, $\alpha$, $\beta$, $A_0$, $V_p$, $V_t$ and $B$. Initial values were specified as xIn = [0.5, 0.1, 0.05, 0.01, 0.1, 1.0, 12.0, 4.0, 1.0, 0.3, 0.3, 0.05]. Boundary conditions were set to 5*xIn for the upper bound and xIn/5 for the lower bound. For fitting, $E_m$ and $\Delta\Psi_m$ were defined as positive voltages to keep all variables non-negative. The residuals were weighted according to Poisson noise, and a safe ODE solver (ode23s) was used with automatic failure handling. Parameters were log transformed to enforce positivity and valid volume fractions. All data were included in the fitting without any weighting factors at different timepoints; however, this could also be included in future iterations of the model. Model performance and fitting errors were evaluated by calculating the coefficient of variance, covariance, correlation matrix and Jacobian returned

by *lsqcurvefit* available in the optimization toolbox of MATLAB (MathWorks Inc.). Least squares estimates were then used as initial conditions for the subsequent MCMC parameter estimation procedure.

### MCMC parameter estimation

Bayesian inference was carried out using an adaptive Metropolis MCMC algorithm, in which the proposal covariance adapts to the posterior distribution during an initial burn-in phase and remains fixed thereafter. A Poisson log-likelihood was used together with weak Gaussian priors. At each step, a new parameter set was generated by adding a normally distributed random number scaled by a factor that determined the step size (sensitivity) of the random walk. Three independent chains with 10,000 iterations were run from different starting points and thinned to reduce autocorrelation. Posterior samples were transformed back to physical parameter values, from which the posterior mean, median, SD and maximum *a posteriori* estimates were calculated. The initial burn-in periods of 1000 iterations were discarded, after which the remaining samples were used to assess convergence of the MCMC iterations using Gelman–Rubin statistics ($\hat{R} \approx 1$). Parameter histograms were constructed from the three trajectories that are centred at the same mean if the trajectories converge to a stable solution. Mean ± SD values were calculated across the remaining 27,000 iterations to provide best-fit estimates and associated standard deviations. Experimental repeats were performed to assess biological variability. Errors are reported as the SD = $\sqrt{S_{\text{flt}}^2 + S_{\text{expt}}^2}$ where $S_{\text{flt}}^2$ and $S_{\text{expt}}^2$ are the mean variances from the fitting and experimental repeats, respectively.

### Data, materials, and software availability

All study data are included in the article. MATLAB code for the fitting procedure is given in the Appendix.

### Results

A representative fit for a control heart ([K$^+$] = 4.9 mM) is shown in Fig. 3*A* acquired with the $\gamma$-detector system following bolus injection of [$^{99m}$Tc]sestamibi into the perfused rat heart, equivalent to constant coronary flow delivery. The best fit is shown in cyan and individual compartment contributions indicated by black, magenta and green lines. The correlation matrix for the derived parameters is shown in Fig. 3*B* with both positively and negatively correlated parameters. As expected, a strong negative correlation is seen between $k_f$ and $k_1$. The three MCMC trajectories with 10,000 iterations for $E_m$ and

$\Delta\Psi_m$ are shown in Fig. 3*C*. Following the initial burn-in period, the chain converged to the same stable equilibrium for the three chains. Corresponding histograms are shown in Fig. 3*D* from which the mean ± SD were calculated.

To evaluate robustness and error behaviour, in particular the strong correlation between flow rate constant $k_f$ and uptake rate constant $k_1$, time activity curves were simulated from eqns (4–8) with the same temporal resolution as the experimental data and known (true) parameters, Fig S1. The input function was modelled as a gamma variate function with parameters $A_0 = 1$, $\alpha = 12$, $\beta = 4$, $t_0 = 1$, assuming zero signal at $t < t_0$. Rate constants were set to $k_1 = 0.1$ and $k_2 = 0.01$ with compartment volume fractions $V_p = 0.3$ and $V_p = 0.3$. Random Poisson noise was added and the fitting parameter $B$ was fixed at zero. Data were simulated with $E_m = -50$ mV and $\Delta\Psi_m = -100$ mV and a range of rate constants for the flow rate constant covering the range $k_f = 0.01$ to 2.0 s$^{-1}$ shown in the Supporting information (Fig. S1*A–F*). Simulations covered the range from slow to fast flow rates with respect to the uptake rate constant $k_1$. The full MCMC procedure was then performed on the simulated data to assess error and bias in the estimated parameter values for $E_m$ and $\Delta\Psi_m$ in the Supporting information (Fig S1*G* and *H*). The fitted curves (see Supporting information, Fig S1*A–F*, solid cyan lines) closely matched the simulated datasets at all flow rates. However, a significant bias and much larger error is observed in the estimated parameters at low flow rates compared to the true values.

The method was next applied to experimental time-activity curves in hearts perfused with normal KHB ([K$^+$] = 4.9 mM) and across a series of hyperkalaemic buffers with [K$^+$] = 10, 15, 20 and 25 mM. A representative fit from a heart perfused with high [K$^+$] (25 mM) is shown in Fig. 4*A* with best fit in cyan and individual compartment contributions indicated by black, magenta and green lines. The three MCMC trajectories with 10,000 iterations for $E_m$ and $\Delta\Psi_m$ are shown in Fig. 4*B* with corresponding histograms in Fig. 4*C*. Contractile function is shown in Fig. 4*D*. Hearts ceased contraction when [K$^+$] exceeded 10 mM, with perfusion maintained at 14 mL min$^{-1}$ in all hearts. The corresponding parameters derived from the MCMC procedure are shown in Fig 4*E* and *F*. Under control conditions, membrane potentials were $E_m = -66 \pm 7$ mV and $\Delta\Psi_m = -109 \pm 9$ mV ($n = 4$, mean ± SD). Summary data across the full [K$^+$] range are shown in Fig. 4*E*. Compared to control, the sarcolemmal membrane potential was depolarized (less negative voltage) in a dose dependent fashion with increasing [K$^+$], whereas $\Delta\Psi_m$ became slightly hyperpolarized (more negative voltage) when the hearts were arrested. Plotting $E_m$ against [K$^+$] (Fig. 4*F*) revealed good agreement with both the GHK (continuous line) and Nernst (dashed line) equations when [K$^+$] was greater

than 10 mM (i.e. in arrested hearts); however, when measured in contracting hearts at more physiological $[K^+]$ (4.9–10 mM), $E_m$ was slightly depolarized compared to theoretical estimates.

To assess mitochondrial contributions to the pharmacokinetics, hearts were perfused with the ionophore CCCP for 10 min to depolarize mitochondria prior to $[^{99m}Tc]$sestamibi or $[^{99m}Tc]$tetrofosmin injection. A representative fit to $[^{99m}Tc]$sestamibi at 600 nM CCCP is shown in Fig. 5A with best fit in cyan and individual compartment contributions indicated by black, magenta and green lines. The three MCMC trajectories with 10,000 iterations for $E_m$ and $\Delta\Psi_m$ are shown in Fig. 5B with corresponding histograms in Fig. 5C. Contractile function is shown in Fig. 5D, which was significantly decreased at 300 nM CCCP and fully arrested at 600 nM, with perfusion maintained at 14 mL min$^{-1}$ in all hearts. Group mean data are summarized in Fig. 5E: mitochondrial potentials were depolarized to $\Delta\Psi_m = -66 \pm 8$ mV ($n = 3$, mean $\pm$ SD) with 300 nM CCCP and to $\Delta\Psi_m = -6 \pm 2$ mV

($n = 3$, mean $\pm$ SD) with 600 nM CCCP. For the highest concentration of 600 nM CCCP, there was almost complete washout of $[^{99m}Tc]$sestamibi. Sarcolemmal potential ($E_m$) was unchanged at 300 nM and only slightly depolarized at 600 nM CCCP. When treated with 300 nM CCCP + 25 mM $[K^+]$ Fig. 5E, both membrane potentials were depolarized to $E_m = -47 \pm 4$ mV ($n = 3$, mean $\pm$ SD) and $\Delta\Psi_m = -78 \pm 5$ mV ($n = 3$, mean $\pm$ SD). Time-activity curves measured with $[^{99m}Tc]$tetrofosmin, also a lipophilic cationic SPECT tracer, were similar to those measured with sestamibi (representative data and MCMC trajectories are shown in the Supporting information, Fig. S2). Mean values of $E_m$ and $\Delta\Psi_m$ measured with $[^{99m}Tc]$tetrofosmin are shown in Fig. 5F. Sarcolemmal membrane potential was $E_m = -59 \pm 6$ mV under control conditions, $E_m = -53 \pm 4$ mV at 300 nM CCCP and $E_m = -51 \pm 5$ mV at 600 nM CCCP ($n = 3$, mean $\pm$ SD), in agreement with those measured by sestamibi. Mitochondrial membrane potential was $\Delta\Psi_m = -103 \pm 10$ mV under control

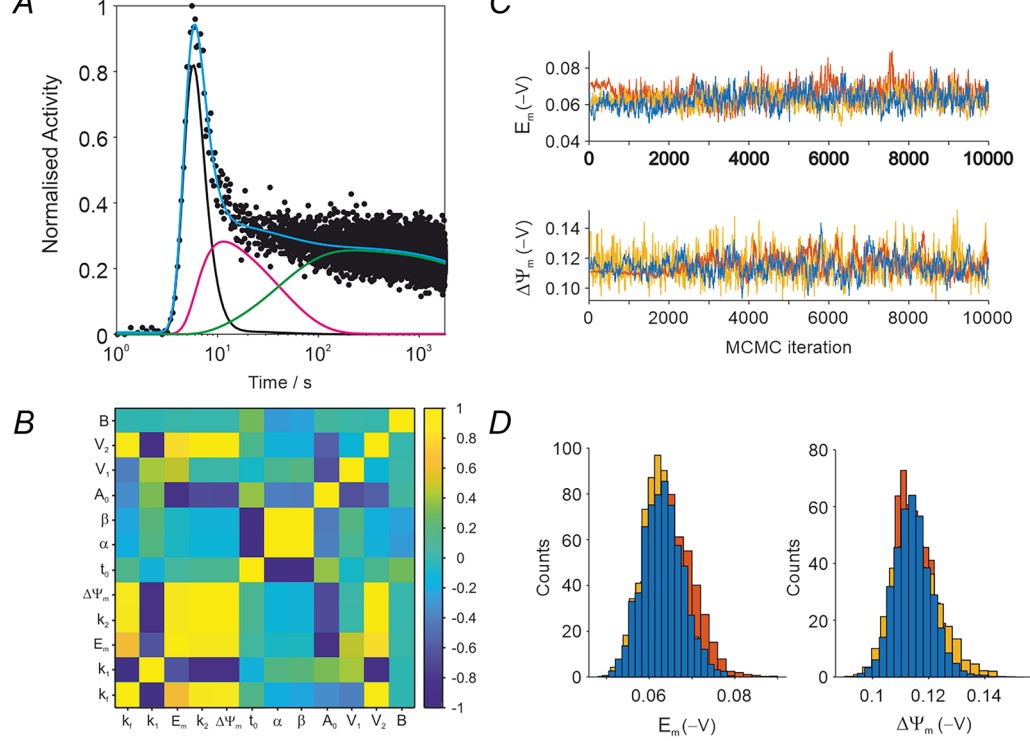

**Figure 3. Example fitting for *n* = 1 control dataset**
*A*, representative time-activity curve following bolus injection of $[^{99m}Tc]$sestamibi into a control rat heart perfused with Krebs–Henseleit buffer (KHB). Initial least squares fit to experimental dataset. Compartment concentrations are shown in black, magenta and green; the cyan line indicates the best fit. The best fit parameters are used as the initial input vector for the MCMC procedure. *B*, correlation matrix of the fitting parameters showing parameters that are positively or negatively correlated. *C*, representative MCMC random walk of the experimental data in (*A*) showing three independent trajectories (10,000 iterations) for the sarcolemmal ($E_m$) and mitochondrial ($\Delta\Psi_m$) membrane potentials with a different starting vector to assess parameter convergence. *D*, histograms of the MCMC trajectories in (*C*). If the chains converge to a stable equilibrium, then the histograms are centred at the same mean value. The burn in period of 1000 datapoints is discarded and the mean $\pm$ SD of the MCMC fitting calculated for the remaining 27,000 iterations. This corresponds to *n* = 1 with associated fitting error.

conditions, $\Delta\Psi_m = -68 \pm 8$ mV at 300 nм CCCP and $\Delta\Psi_m = -9 \pm 3$ mV at 600 nм CCCP ($n = 3$, mean $\pm$ SD), also in agreement with those measured by sestamibi. For the highest concentration of 600 nм CCCP, there was almost complete washout of [$^{99m}$Tc]tetrofosmin.

Dynamic planar scintigraphy data are illustrated in Fig. 6*A*, with a ROI drawn over the heart. A representative static image acquired 2 h post injection is shown in Fig. 6*B*. The corresponding time–activity curve is displayed in Fig. 6*C*, showing some qualitative differences compared to perfused heart experiments, since the plasma compartment also incorporates ventricular blood volume *in vivo*. The three MCMC trajectories with 10,000 iterations for $E_m$ and $\Delta\Psi_m$ are shown in Fig. 6*D* with corresponding histograms in Fig. 6*E*. Group mean estimates of membrane potentials are summarized in Fig. 6*F*. Estimated membrane potentials *in vivo* were $E_m = -61 \pm 8$ mV and $\Delta\Psi_m = -151 \pm 13$ mV ($n = 4$, mean $\pm$ SD).

## Discussion

We developed a modelling approach for the kinetics of [$^{99m}$Tc]sestamibi that allows non-invasive estimation of both sarcolemmal and mitochondrial membrane potentials in the heart both *ex vivo* and *in vivo*. This approach using subnanomolar concentrations of radiotracer is not subject to the toxicity and concentration-dependent artefacts associated with more widely used optical techniques with fluorophores (Kowaltowski & Abdulkader, 2024) and, unlike [$^{3}$H]-based or mass spectrometry-based approaches (Logan et al., 2016), do not require biopsy and can be performed serially in the same tissue or individual over time. It is also tissue depth independent, potentially whole body, and, by repurposing clinically established and widespread radiotracers such as [$^{99m}$Tc]sestamibi and [$^{99m}$Tc]tetrafosmin, it has potential as both an experimental and diagnostic tool cross-translatable

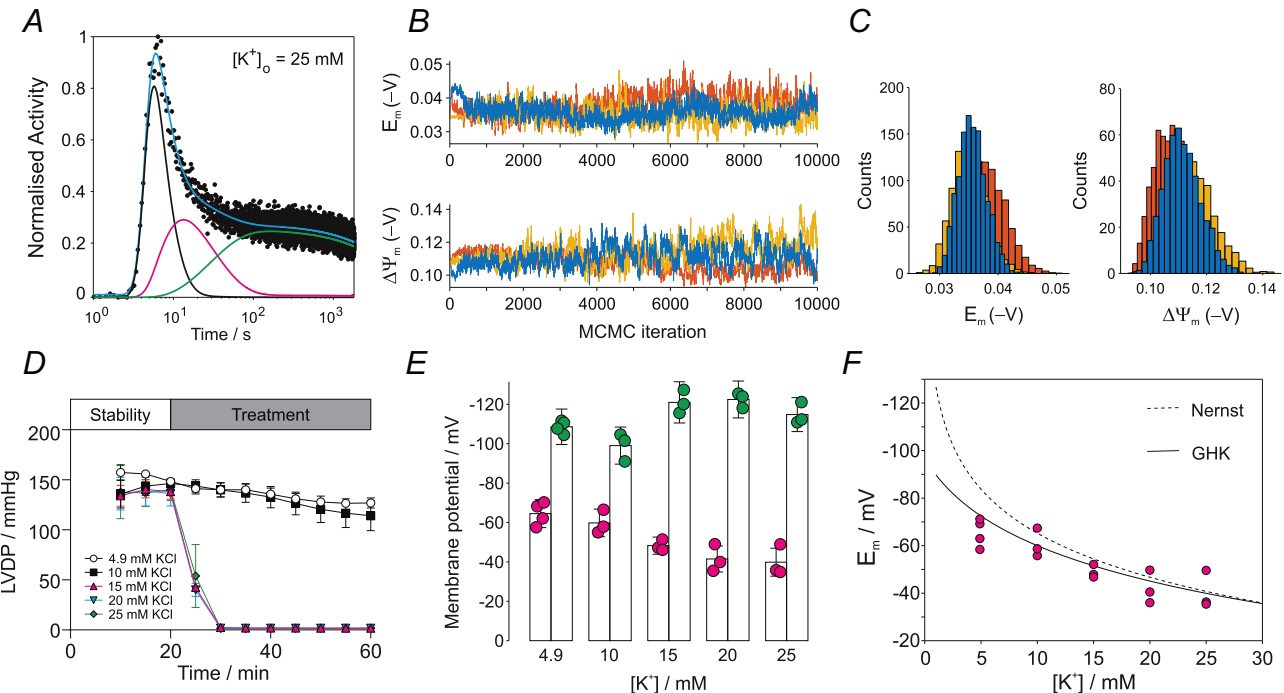

**Figure 4. Effect of extracellular potassium concentration on sarcolemmal and mitochondrial membrane potentials**

*A*, representative time-activity curve following bolus injection of [$^{99m}$Tc]sestamibi into rat hearts perfused with Krebs–Henseleit buffer (KHB) containing 25 mм K$^+$. The cyan line shows the best model fit; individual compartments are plasma ($c_p(t)$, black), cytosol ($c_c(t)$, magenta) and mitochondria ($c_m(t)$, green). *B*, MCMC random walk of the experimental data in (*A*) showing three independent trajectories (10,000 iterations) for the sarcolemmal ($E_m$) and mitochondrial ($\Delta\Psi_m$) membrane potentials. *C*, histograms of the MCMC trajectories in (*B*). *D*, left ventricular developed pressure (LVDP) in hearts perfused with control KHB and increasing [K$^+$]. Data are presented as the mean $\pm$ SD. *E*, mean values of sarcolemmal ($E_m$) and mitochondrial ($\Delta\Psi_m$) membrane potentials derived from MCMC fitting in control ($n = 4$) and those perfused with hyperkaliaemic buffers ($n = 3$). Data are presented as the mean $\pm$ SD (see text for error estimate). *F*, values of $E_m$ from (*E*) replotted against extracellular [K$^+$], compared with theoretical predictions from the Goldman–Hodgkin–Katz equation (continuous line) and the Nernst equation for the K$^+$ equilibrium voltage $E_K$ (dashed line).

between rodent models of disease (Safee et al., 2019) and the clinic.

In the isolated perfused heart model, titration with hyperkalaemic buffers led to a dose-dependent decrease in the estimated $E_m$ with increasing $[K^+]$, agreeing well with the predictions made using the GHK and Nernst equations when $[K^+]$ was greater than 10 mM and the hearts were arrested. However, the estimated value of $E_m = -66 \pm 7$ mV in beating control hearts when $[K^+] = 4.9$ mM was less negative than predicted by the models and lower than previously reported values of $E_m = -70$ mV during cold ischaemic arrest (Snabaitis et al., 1997). In a beating heart, the action potential duration $APD_{50}$ is ~30 ms in the rat (Schouten, 1984), representing the plateau depolarization phase, whereas the duration of the cardiac cycle is 200 ms for a heart rate of 300 beats $min^{-1}$. Assuming a diastolic membrane potential of $E_m = -80$ mV for a duration of 170 ms and a systolic membrane potential of $E_m = +40$ mV lasting 30 ms, the time-averaged voltage over the entire cardiac cycle is ~$E_m = -62$ mV. Our measurements of the sarcolemmal membrane potential in control hearts *ex vivo* (which were in agreement with those measured *in vivo*) probably reflect this time-averaged voltage over the cardiac cycle, whereas arrested hearts are closer to the model predictions.

Treatment of hearts with CCCP led to significantly increased washout rates of both tracers. Almost complete washout of the tracers at the highest concentration of 600 nM CCCP suggests that there is no non-specific binding that would be a confounding factor in the estimation of membrane potentials. It is of interest that the sarcolemmal membrane potential remained polarized during CCCP treatment despite mitochondrial ATP production having probably ceased under this intervention. This is consistent with the hypothesis that sarcolemmal ionic homeostasis is primarily supported by glycolytic ATP (independent of mitochondrial ATP), facilitated by the physical and functional association of sarcolemmal ion pumps and channels with sub-

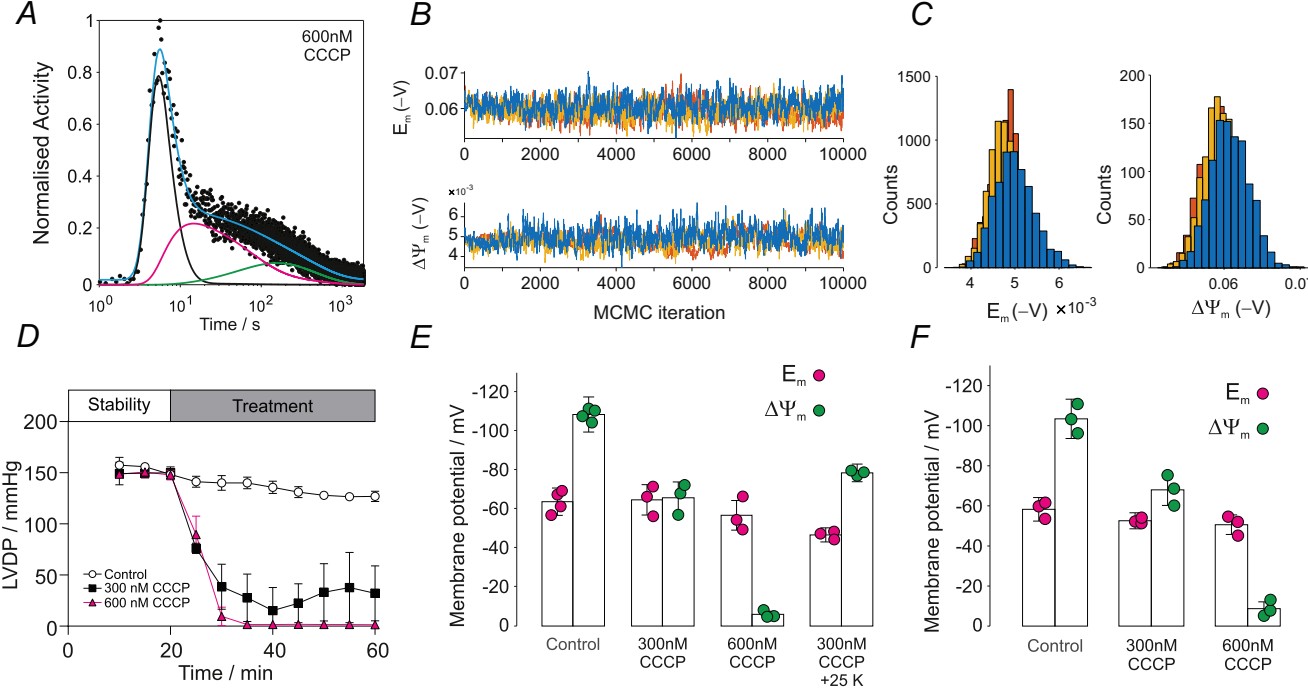

**Figure 5. Effect of mitochondrial depolarization with CCCP on sarcolemmal and mitochondrial membrane potentials**

*A*, representative time activity curve following bolus injection of [$^{99m}$Tc]sestamibi into rat hearts perfused with Krebs–Henseleit buffer (KHB) containing 600 nM CCCP. The cyan line shows the best model fit; individual compartments are plasma ($c_p(t)$, black), cytosol ($c_c(t)$, magenta) and mitochondria ($c_m(t)$, green). *B*, MCMC random walk of the experimental data in (*A*) showing three independent trajectories (10,000 iterations) for the sarcolemmal ($E_m$) and mitochondrial ($\Delta\Psi_m$) membrane potentials. *C*, histograms of the MCMC trajectories in (*B*). *D*, left ventricular developed pressure (LVDP) in hearts perfused with control KHB, 300 nM CCCP and 600 nM CCCP. Data are presented as the mean $\pm$ SD. *E*, mean sarcolemmal ($E_m$) and mitochondrial ($\Delta\Psi_m$) potentials derived from MCMC fitting of [$^{99m}$Tc]sestamibi in control (*n* = 4), 300 nM CCCP (*n* = 3), 600 nM CCCP (*n* = 3) and combined 300 nM CCCP + 25 mM K$^+$ (*n* = 3) conditions. *F*, mean $E_m$ and $\Delta\Psi_m$ values obtained with [$^{99m}$Tc]tetrofosmin in control (*n* = 3), 300 nM CCCP (*n* = 3) and 600 nM CCCP (*n* = 3) hearts. Data are presented as the mean $\pm$ SD (see text for detail of error estimate).

sarcolemmal glycolytic complexes, as previously described in many tissue types including but not limited to the heart (Balaban & Bader, 1984; Dhar-Chowdhury et al., 2007; Hong et al., 2011; Meyer et al., 2022).

Our estimated mitochondrial membrane potential $\Delta\Psi_m = -109 \pm 9$ mV ($n = 4$, mean $\pm$ SD) in the Langendorff perfused heart *ex vivo* agrees with those reported in the literature measured with [$^3$H]TPMP ($-125 \pm 7$ mV) (Kauppinen, 1983) and [$^3$H]TPP$^+$ in glucose only perfused rat hearts ($\Delta\Psi_m = -118.2 \pm 1.4$, $-108.0 \pm 1.5$ and $-100.8 \pm 1.0$ mV at low, medium and high workloads, respectively) (Wan et al., 1993). When perfused with a more physiological range of substrates we found $\Delta\Psi_m$ to be $-20$ mV more polarized (Hoare et al., 2025). However, our *in vivo* measurements

of mitochondrial membrane potential ($\Delta\Psi_m = -151 \pm 13$ mV ($n = 4$, mean $\pm$ SD) were substantially more negative than those measured *ex vivo*, and closer to accepted values, suggesting that the lower voltages measured in the perfused heart may reflect suboptimal metabolism of a glucose-only crystalloid perfused heart that lacks the full range of substrates that would be available *in vivo*, or energetic differences between the mechanically unloaded Langendorff perfused heart and the heart *in vivo*, which is better oxygenated but performing active work (Southworth et al., 2005; Sutherland & Hearse, 2000).

Although we consider the present study to be the first to distinguish mitochondrial from sarcolemmal membrane potentials using radiometric methods, previous

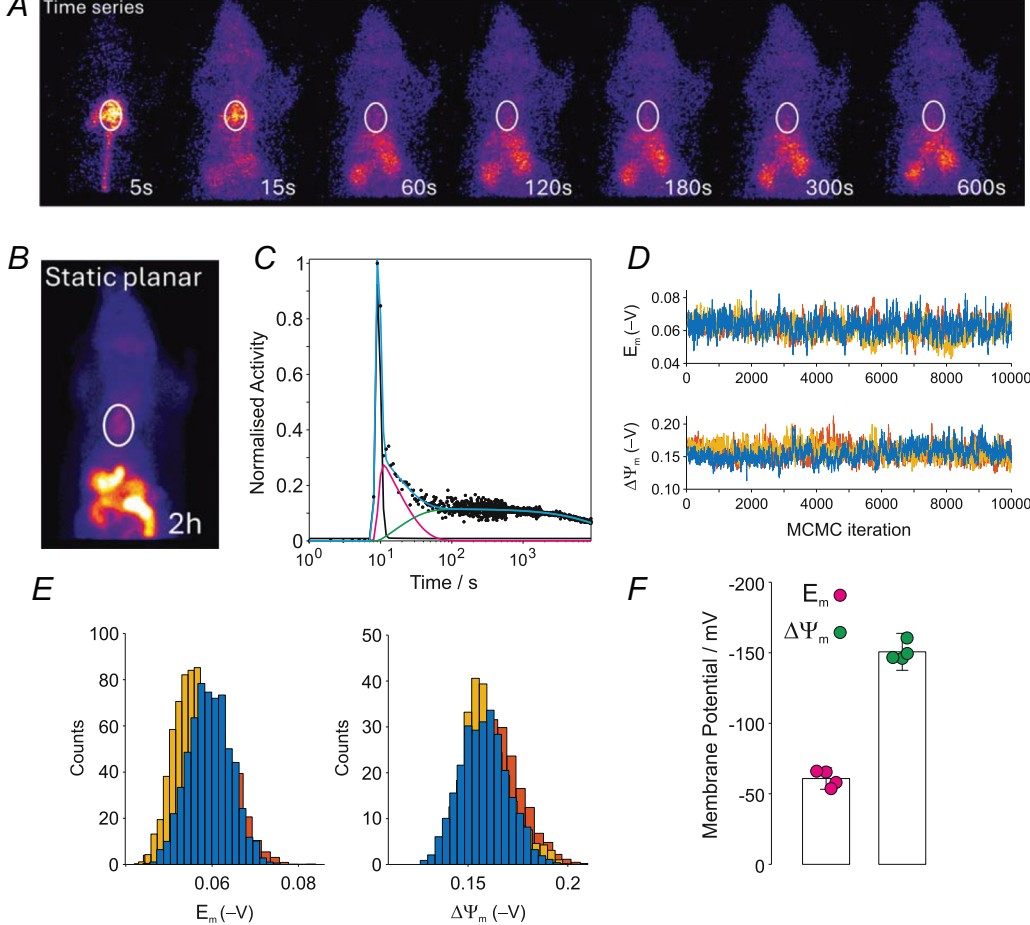

**Figure 6. In vivo estimation of sarcolemmal and mitochondrial membrane potentials**
*A*, dynamic time-series of planar SPECT scintigraphy images following bolus injection of [$^{99m}$Tc]sestamibi in a healthy anaesthetized rat, with a region of interest (ROI) placed over the heart. *B*, static planar SPECT image acquired over 10 min at 2.5 h post injection. *C*, representative time-activity curve for the heart ROI. The cyan line shows the best model fit; plasma ($c_p(t)$, black), cytosol ($c_c(t)$, magenta) and mitochondria ($c_m(t)$, green) are shown separately. *D*, MCMC random walk of the experimental data in (*C*) showing three independent trajectories (10,000 iterations) for the sarcolemmal ($E_m$) and mitochondrial ($\Delta\Psi_m$) membrane potentials. *E*, histograms of the MCMC trajectories in (*D*). *F*, mean sarcolemmal and mitochondrial membrane potentials measured *in vivo* were $E_m = -61 \pm 8$ mV and $\Delta\Psi_m = -151 \pm 13$ mV, respectively ($n = 4$). Data are presented as the mean $\pm$ SD (see text for detail of error estimate).

measurements of total myocardial membrane potential report $E_t = -148.1 \pm 6$ mV in dogs, $-146.7 \pm 3.8$ mV in rats and $-139.3 \pm 5.8$ mV in mice using $[^{11}C]TPMP$ (Fukuda et al., 1986), whereas $E_t = -160.7 \pm 3.7$ mV was reported in humans using $[^{18}F]TPP$ (Pelletier-Galarneau et al., 2021). These measurements were performed by continuous infusion of tracer that was allowed to reach a steady state and the tracer distribution modelled as a two-compartment system where the tissue compartment was subject to a single voltage $E_t = E_m + \Delta\Psi_m$. The extracellular volume fraction was measured by contrast enhanced MRI with blood sampling to measure plasma radioactivity. Our measurements would give $E_t = -213$ mV, which does not agree with these previous measurements, the reported values of total $E_t$ are in better agreement with our values of $\Delta\Psi_m$. This can be rationalized by noting that voltages of $-66$ mV and $-151$ mV give a calculated equilibrium constant of $K_1 = 12$ across the sarcolemmal membrane and $K_2 = 285$ across the mitochondrial membrane. At these voltages, the relative steady-state concentrations in the plasma : cytosol : mitochondria are 1 : 12 : 3420, respectively. The plasma and cytosolic concentrations would therefore be largely negligible compared to that in the mitochondria which would therefore inappropriately dominate estimates of $E_t$.

Methods such as the one proposed here, as well as others in the literature (Alpert et al., 2018; Pelletier–Galarneau et al., 2021), have great potential to inform clinical end-points. Assessing myocardial viability has been an important application in clinical cardiovascular MRI. More generally mitochondrial dysfunction is a key component of cardiovascular pathologies, as well as many other disease aetiologies. Evidence also points to the emerging role that mitochondrial dysfunction plays in cardiotoxicity in oncology patients during or after treatment with chemotherapies.

### Study limitations and future directions

A limitation of the present study is the unavoidable use of anaesthetics in preclinical studies. Highly lipophilic anaesthetics such as pentobarbital and isoflurane perturb mitochondrial $\Delta\Psi_m$ by altering bioenergetics (e.g. complex I inhibition, increased proton/$K^+$ conductance, mitoKATP channel activation). Although this would not affect our measurements of sarcolemmal membrane potentials, it is possible that the use of anaesthetics causes a depolarization of the mitochondrial membrane potential which would lead to an underestimation of the voltages.

Although the approach yields an effective volume and time-averaged $E_m$ and $\Delta\Psi_m$, heterogeneity in mitochondrial populations, regional perfusion or trans-porter expression could bias estimates or broaden posteriors. $[^{99m}Tc]$sestamibi is a known substrate for P-gp, an ATP dependent efflux pump that is important for the biodistribution of pharmaceutical agents, assessment of liver and kidney toxicity and contributes to drug resistance (e.g. in cancer). P-gp is also a major component of the blood–brain barrier and therefore it is interesting to note that brain uptake of $[^{99m}Tc]$sestamibi seen in Fig. 6 is low. High levels of P-gp expression would pump $[^{99m}Tc]$sestamibi out of the cell leading to increased washout rates. A major application of $[^{99m}Tc]$sestamibi in the clinic is for this very purpose. Therefore, high expression levels of P-gp in certain tissues such as liver, kidney or in some cancers may confound the measurement of membrane potentials in certain tissues and organs. Expression levels are much lower in the heart. The data presented here strongly supports that the mechanism for retention and pharmacokinetics of $[^{99m}Tc]$sestamibi and $[^{99m}Tc]$tetrofosmin in the heart are determined by the membrane voltages. In principle, the framework is transferable to other organs and disease contexts provided tracer perfusion is adequate and organ-specific confounds are accounted for such as blood–brain barrier/low uptake, higher P-gp expression, different extracellular volumes and the need for 3D dynamic PET/SPECT for regional mapping. Further experiments will be required to determine the importance of P-gp expression on the pharmacokinetics. Possible deviations from ideal Nernst behaviour are also possible where molar activities do not equate to concentration, particularly non-specific binding, which would perturb the equilibrium distribution across the membrane.

Possible (generic) limitations are also associated with the complexity of the MCMC fitting method; and possible correlation of the fitted variables can arise. As is usual with this methodology, cross-correlation was assessed on simulated data (using the model of the system) by calculating a correlation matrix, coefficients of variance and the fitting Jacobian. Fixing the values of some of the parameters, which might be estimated in separate experiments, would reduce the variances of the fitted values.

Another challenge arose in our analysis as a result of the large size of the data files (*ex vivo* data were acquired with ∼9000 time points, whereas *in vivo* data were acquired with 1260 data points) with their very high temporal resolution slowing down the parameter fitting by the MCMC method. Such high temporal resolution was achieved by using gamma detectors and planar scintigraphy, which is not possible with 3D SPECT imaging due to rotation of the camera. Planar scintigraphy has a further limitation in that it offers no depth resolution. It would therefore be challenging to assess regional changes in membrane potential using this technique. This limitation is mitigated with PET

imaging, which is natively dynamic with high temporal 3D resolution, and for which we are currently developing appropriate new lipophilic cations (McCluskey et al., 2019; Osborne et al., 2021; Smith et al., 2018) including a PET-compatible approach for sestamibi radiolabelled with $^{94m}$Tc (Harper et al., 2024).

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

## Additional information

### Data availability statement

All of the data supporting the results presented in the published article are included in the figures.

## Competing interests

The authors declare that they have no competing interest.

## Author contributions

E.C.T.W., M.J.S., R.S. and T.R.E. designed research. E.C.T.W. and F.B. performed experiments. M.R.O. and T.R.E. developed theory and wrote the MATLAB code. T.R.E. analyzed data. R.S. and T.R.E. wrote the paper. All authors approved the final version of the manuscript submitted for publication.

## Funding

The School of Biomedical Engineering and Imaging Sciences is supported by the Wellcome EPSRC Centre for Medical Engineering at King's College London (WT 203148/Z/16/Z) and the Department of Health via the National Institute for Health Research (NIHR) comprehensive Biomedical Research Centre award to Guy's & St Thomas' NHS Foundation Trust in partnership with King's College London and King's College Hospital NHS Foundation Trust. This work was supported by the EPSRC Programme grants EP/S032789/1 and EP/S019901/1; British Heart Foundation Programme Grants RG/12/4/29426 and RG/17/15/33106 and the BHF Centre of Research Excellence RE/24/130035.

## Acknowledgements

We thank Professor Philip W. Kuchel, Dr Joel T. Dunn and Professor Alexander Hammers for critically reading the manuscript and PWK for advising on the use of Markov chain Monte Carlo methods for parameter estimation by systematic random sampling of high-dimensional probability distributions.

## Keywords

heart, mitochondrial dysfunction, mitochondrial membrane potential, sarcolemmal membrane potential, [$^{99m}$Tc]sestamibi pharmacokinetics, technetium-labelled membrane potential probe

## Supporting information

Additional supporting information can be found online in the Supporting Information section at the end of the HTML view of the article. Supporting information files available:

**Peer Review History**
**Supplementary Information**

