## [Peer Review History · The Journal of Physiology]

Sarcolemmal and mitochondrial membrane potentials measured *ex vivo* and *in vivo* in the heart by pharmacokinetic modelling of [^{99m}Tc]sestamibi

Edward C.T. Waters, Friedrich Baark, Matthew R. Orton, Michael J Shattock, Richard Southworth, and Thomas R. Eykyn
DOI: 10.1113/JP290295

Corresponding author(s): Thomas Eykyn (thomas.eykyn@kcl.ac.uk)

The following individual(s) involved in review of this submission have agreed to reveal their identity: Ken D O'Halloran (Referee #1); Jin Han (Referee #2); Frank R Heinzl (Referee #3)

Review Timeline:

Submission Date:	09-Oct-2025
Editorial Decision:	06-Nov-2025
Revision Received:	29-Dec-2025
Editorial Decision:	21-Jan-2026
Revision Received:	26-Jan-2026
Accepted:	09-Feb-2026

Senior Editor: Natalia Trayanova

Reviewing Editor: Brian Delisle

Transaction Report:

Re: JP-RP-2025-290295 "Sarcolemmal and mitochondrial membrane potentials measured *ex vivo* and *in vivo* in the heart by pharmacokinetic modelling of [^{99m}Tc]sestamibi" by Edward C.T. Waters, Friedrich Baark, Matthew R. Orton, Michael J Shattock, Richard Southworth, and Thomas R. Eykyn

Dear Dr Eykyn,

Thank you for submitting your manuscript to The Journal of Physiology. It has been assessed by a Reviewing Editor and by 3 expert referees and we are pleased to tell you that it is potentially acceptable for publication following satisfactory major revision.

Please address all the points raised and incorporate all requested revisions or explain in your Response to Referees why a change has not been made. We hope you will find the comments helpful and that you will be able to return your revised manuscript within 9 months. If you require an extension, please contact journal staff: jp@physoc.org. Please note that this letter does not constitute a guarantee for acceptance of your revised manuscript.

LANGUAGE EDITING AND SUPPORT FOR PUBLICATION: If you would like help with English language editing, or other article preparation support, Wiley Editing Services offers expert help, including English Language Editing, as well as translation, manuscript formatting, and figure formatting at www.wileyauthors.com/eoo/preparation. You can also find resources for Preparing Your Article for general guidance about writing and preparing your manuscript at www.wileyauthors.com/eoo/prepresources.

REVISION CHECKLIST:

We look forward to receiving your revised submission.

Yours sincerely,

Natalia Trayanova
Senior Editor
The Journal of Physiology

REQUIRED ITEMS

- You must start the Methods section with a paragraph headed Ethical approval (https://jp.msubmit.net/cgi-bin/main.plex?form_type=display_requirements#methods).

Research must comply with The Journal's policies regarding animal experiments (<https://physoc.onlinelibrary.wiley.com/hub/animal-experiments>) and adherence to these policies must be stated in the manuscript.

Authors should confirm in their Methods section that their experiments were carried out according to the guidelines laid down by their institution's animal welfare committee, including an ethics approval reference number. The Methods section must contain a statement about access to food, water and housing, details of the anaesthetic regime: anaesthetic used, dose and route of administration, and method of killing the experimental animals.

- The reference list must be in alphabetical order, rather than numbered, to comply with our Journal format.

- Please ensure that the Article File you upload is a Word file.

- Please include an Abstract Figure file and an Abstract Figure legend. An appropriate figure legend, which should not exceed 150 words in length, should be included in the main manuscript file. The Abstract Figure is a piece of artwork designed to give readers an immediate understanding of the research and should summarise the main conclusions. If possible, the image should be easily 'readable' from left to right or top to bottom. It should show the physiological relevance of the manuscript so readers can assess the importance and content of its findings. Abstract Figures should not merely recapitulate other figures in the manuscript. Please try to keep the diagram as simple as possible and without superfluous information that may distract from the main conclusion(s). Abstract Figures must be provided by authors no later than the revised manuscript stage and should be uploaded as a separate file during online submission labelled as File Type 'Abstract Figure'. Please also ensure that you include the figure legend in the main article file. All Abstract Figures should be created using BioRender. Authors should use The Journal's premium BioRender account to export high-resolution images. Details on how to use and access the premium account are included as part of this email.

EDITOR COMMENTS

Reviewing Editor:

Three expert referees in the field have reviewed the work. Both referees agreed that a revised manuscript has the potential to significantly impact the field. They each identified concerns that, if addressed, would improve the overall quality and clarity of the work. In addition, the ethics editor identified several points that need to be identified to conform with journal policy.

Please also see 'Required Items' above.

REFEREE COMMENTS

Referee #1:

Thank you for submitting your manuscript to The Journal of Physiology. Some additional details pertaining to animal ethics and welfare are required.

1. You must start the methods section with the subheading "Ethical approval". Include the institutional approval code/number for the study.

2. Please add text to confirm that the study conformed to the ethical principles of The Journal of Physiology.

3. Provide details on the source of the animals. Briefly describe husbandry arrangements and confirm in the text that animals had free access to food and water.

4. Please state killed (or euthanised) by intraperitoneal injection of....

State how death was confirmed.

5. Please note that there are growing concerns with the use of pentobarbitone even for single terminal use as it is an irritant. I note that its use was authorised in the study by the relevant bodies, and therefore is fully acceptable, but The Journal is currently encouraging authors to consider more refined approaches for euthanasia for future work, unless there are strong grounds for justification of use and/or steps to refine its use, such as attendant use of lidocaine.

Referee #2:

1. Please clarify novelty in plain terms. Explicitly define in the Abstract or Introduction that "this is the first method to independently and quantitatively resolve sarcolemmal and mitochondrial membrane potentials in vivo using dynamic radiotracer modelling."

2. Provide a succinct paragraph discussing potential deviations from the ideal Nernst equilibrium (e.g., non-homogeneous mitochondrial populations, active transporters, or changes in perfusion) and their impact on $\Delta\Psi_m$ estimation.

3. The Discussion should include 2-3 sentences on how this framework could be adapted for other tissues (brain, liver) or diseases (diabetes, neurodegeneration), thus underlining its conceptual breadth.

4. While figures are clear, including a schematic summarizing how E_m and $\Delta\Psi_m$ are mathematically derived from tracer kinetics (Equation links \rightarrow model \rightarrow MCMC output) would improve readability for physiologists less familiar with radiokinetic modelling.

5. Throughout the manuscript, ensure consistent notation and sign conventions for potentials (negative vs positive voltages). Please clarify in the Methods that E_m and $\Delta\Psi_m$ are defined as negative inside the membrane for interpretive consistency.

6. The reported SDs (± 1 mV for E_m ex vivo) seem surprisingly low given experimental and modelling variability. Please discuss the possible underestimation of uncertainty due to correlated parameters or limited sample size.
7. Strengthen the last paragraph of the Discussion by explicitly connecting this technique to clinical endpoints, such as assessing myocardial viability, mitochondrial dysfunction, or early cardiotoxicity in oncology patients.
8. Abstract: Add a one-sentence concluding statement highlighting clinical and physiological implications.
9. Introduction: Replace phrases like "we developed a modelling framework" with "we established a quantitative imaging paradigm," which conveys broader conceptual weight.
10. Discussion: Combine the "Study Limitations" section with a forward-looking "Future Directions" paragraph for smoother flow.
11. References: Include recent work on radiolabeled mitochondrial probes (e.g., 2024-2025 total-body PET studies) to emphasize current relevance.
12. Supplementary Materials: Consider including convergence plots or trace diagnostics for MCMC to reassure readers of parameter stability.

Referee #3:

The authors propose and test a modelling approach to estimate sarcolemmal (E_m) and mitochondrial ($\Delta\Psi_m$) membrane potentials in rat hearts using planar scintigraphy with [^{99m}Tc]sestamibi and [^{99m}Tc]tetrofosmin. By combining first-pass pharmacokinetics, the Nernst equation, and Markov Chain Monte Carlo fitting, they derive time-averaged potentials ex vivo in Langendorff-perfused hearts and in vivo, and compare these values with published data. The approach is conceptually elegant and in particular from the modelling perspective methodologically innovative. The following comments aim to help clarify specific methodological aspects and strengthen the interpretation and translational scope.

RC#1: The definition of the "first-pass" period used for kinetic modelling remains ambiguous. In Fig. 4, model fits extend to ~ 1000 s, suggesting that data beyond the first-pass transit may have been included in parameter estimation. Because tracer retention after initial uptake is influenced by mitochondrial binding and slow washout rather than instantaneous membrane potentials, the authors should clarify a. the exact temporal range of data used for fitting (preferably also indicated in the figure), b. whether later time points were weighted or excluded, and c. how this choice affects the interpretation of E_m and $\Delta\Psi_m$ as first-pass rather than steady-state parameters.

RC#2: The authors have appropriately validated their kinetic model using simulated data and examined parameter correlations qualitatively. To strengthen confidence in the robustness of the voltage estimates, it would be valuable to quantify the sensitivity of E_m and $\Delta\Psi_m$ to variations in the fitted transport rate constants (k_1 , k_2). Given that these parameters are influenced by myocardial perfusion and tracer diffusion characteristics, the authors might compare their fitted values with independent perfusion or extraction data for [^{99m}Tc]sestamibi and [^{99m}Tc]tetrofosmin, or report sensitivity coefficients or correlations derived from the Jacobian. This would clarify the stability of the potential estimates against physiologically plausible variations in tracer kinetics.

RC#3: While the study convincingly demonstrates the feasibility of estimating sarcolemmal and mitochondrial membrane potentials in vivo using radiotracer kinetics, the manuscript would benefit from a clearer discussion of the temporal and spatial resolution limitations inherent to the approach. In contrast to optical techniques, which in experimental settings allow continuous and subcellular monitoring of membrane potential dynamics, the presented method provides single, first-pass "snapshot" measurements with millimetre-scale spatial resolution due to planar scintigraphy. Explicitly quantifying these constraints and discussing their implications for detecting rapid or regional potential changes would strengthen the reader's understanding of the technique's current scope and translational potential.

END OF COMMENTS

Reviewer comments:

The authors propose and test a modelling approach to estimate sarcolemmal (E_m) and mitochondrial ($\Delta\Psi_m$) membrane potentials in rat hearts using planar scintigraphy with [^{99m}Tc]sestamibi and [^{99m}Tc]tetrofosmin. By combining first-pass pharmacokinetics, the Nernst equation, and Markov Chain Monte Carlo fitting, they derive time-averaged potentials *ex vivo* in Langendorff-perfused hearts and *in vivo*, and compare these values with published data. The approach is conceptually elegant and in particular from the modelling perspective methodologically innovative. The following comments aim to help clarify specific methodological aspects and strengthen the interpretation and translational scope.

RC#1: The definition of the “first-pass” period used for kinetic modelling remains ambiguous. In Fig. 4, model fits extend to ~1000 s, suggesting that data beyond the first-pass transit may have been included in parameter estimation. Because tracer retention after initial uptake is influenced by mitochondrial binding and slow washout rather than instantaneous membrane potentials, the authors should clarify a. the exact temporal range of data used for fitting (preferably also indicated in the figure), b. whether later time points were weighted or excluded, and c. how this choice affects the interpretation of E_m and $\Delta\Psi_m$ as first-pass rather than steady-state parameters.

RC#2: The authors have appropriately validated their kinetic model using simulated data and examined parameter correlations qualitatively. To strengthen confidence in the robustness of the voltage estimates, it would be valuable to quantify the sensitivity of E_m and $\Delta\Psi_m$ to variations in the fitted transport rate constants (k_1 , k_2). Given that these parameters are influenced by myocardial perfusion and tracer diffusion characteristics, the authors might compare their fitted values with independent perfusion or extraction data for [^{99m}Tc]sestamibi and [^{99m}Tc]tetrofosmin, or report sensitivity coefficients or correlations derived from the Jacobian. This would clarify the stability of the potential estimates against physiologically plausible variations in tracer kinetics.

RC#3: While the study convincingly demonstrates the feasibility of estimating sarcolemmal and mitochondrial membrane potentials *in vivo* using radiotracer kinetics, the manuscript would benefit from a clearer discussion of the temporal and spatial resolution limitations inherent to the approach. In contrast to optical techniques, which in experimental settings allow continuous and subcellular monitoring of membrane potential dynamics, the presented method provides single, first-pass “snapshot” measurements with millimetre-scale spatial resolution due to planar scintigraphy. Explicitly quantifying these constraints and discussing their implications for detecting rapid or regional potential changes would strengthen the reader’s understanding of the technique’s current scope and translational potential.

Dear Editor and Referees,

We are very grateful for your time in reviewing our work and for the thoughtful and helpful reviews. Two independent comments by Referees #2 and #3 necessitated us revisiting our analysis. As a result, we decided to rewrite the MATLAB script to provide a more robust estimation of the parameters and a reanalysis of the associated errors. Consequently, we have reanalysed ALL the data in our manuscript and added additional simulations. The overall results are unchanged, however the technical reproducibility in the parameter estimates are now more appropriate for such multi parametric fitting. Consequently, all figures have been modified and replaced with the new analysis, including two new supplementary figures. Please find below detailed responses to all the Referee comments as well as the Editorial reviews. Our responses are in green text and additions to the manuscript text in red.

Re: JP-RP-2025-290295 "Sarcolemmal and mitochondrial membrane potentials measured ex vivo and in vivo in the heart by pharmacokinetic modelling of [99mTc]sestamibi" by Edward C.T. Waters, Friedrich Baark, Matthew R. Orton, Michael J Shattock, Richard Southworth, and Thomas R. Eykyn

Dear Dr Eykyn,

Thank you for submitting your manuscript to The Journal of Physiology. It has been assessed by a Reviewing Editor and by 3 expert referees and we are pleased to tell you that it is potentially acceptable for publication following satisfactory major revision.

We are very grateful for the overall positive comments and are pleased that you found our work interesting.

Please address all the points raised and incorporate all requested revisions or explain in your Response to Referees why a change has not been made. We hope you will find the comments helpful and that you will be able to return your revised manuscript within 9 months. If you require an extension, please contact journal staff: jp@physoc.org. Please note that this letter does not constitute a guarantee for acceptance of your revised manuscript.

Your revised manuscript should be submitted online using the link in your Author Tasks:

[https://jp.msubmit.net/cgi-](https://jp.msubmit.net/cgi-bin/main.plex?el=A4JS7Hkm2A4UbZ5F3A9ftd7lROJH0XCysW7GfIRNYbJwZ)

[bin/main.plex?el=A4JS7Hkm2A4UbZ5F3A9ftd7lROJH0XCysW7GfIRNYbJwZ](https://jp.msubmit.net/cgi-bin/main.plex?el=A4JS7Hkm2A4UbZ5F3A9ftd7lROJH0XCysW7GfIRNYbJwZ). This link is accessible via your account as Corresponding Author; it is not available to your co-authors. If this presents a problem, please contact journal staff (jp@physoc.org). Image files from the previous version are retained on the system. Please ensure you replace or remove any files that are being revised.

If you do not wish to submit a revised version of your manuscript, you must inform our journal staff (jp@physoc.org) or reply to this email to request withdrawal. Please note that a manuscript must be

formally withdrawn from the peer review process at one journal before it may be submitted to another journal.

LANGUAGE EDITING AND SUPPORT FOR PUBLICATION: If you would like help with English language editing, or other article preparation support, Wiley Editing Services offers expert help, including English Language Editing, as well as translation, manuscript formatting, and figure formatting at www.wileyauthors.com/eo/preparation. You can also find resources for Preparing Your Article for general guidance about writing and preparing your manuscript at www.wileyauthors.com/eo/prepresources.

REVISION CHECKLIST:

We look forward to receiving your revised submission.

Yours sincerely,

Natalia Trayanova

Senior Editor

The Journal of Physiology

REQUIRED ITEMS

- You must start the Methods section with a paragraph headed Ethical approval (https://jp.msubmit.net/cgi-bin/main.plex?form_type=display_requirements#methods).

We have started the Methods section with the requested Ethical approval paragraph.

Research must comply with The Journal's policies regarding animal experiments (<https://physoc.onlinelibrary.wiley.com/hub/animal-experiments>) and adherence to these policies must be stated in the manuscript.

We have added a statement, see comment to reviewer 1.

Authors should confirm in their Methods section that their experiments were carried out according to the guidelines laid down by their institution's animal welfare committee, including an ethics approval reference number. The Methods section must contain a statement about access to food, water and housing, details of the anaesthetic regime: anaesthetic used, dose and route of administration, and method of killing the experimental animals.

Confirmed and additional details added to the Materials and Methods section.

- The reference list must be in alphabetical order, rather than numbered, to comply with our Journal format.

This has been rectified in the revised manuscript. In text citations are now formatted as (Author, year) and are presented in alphabetical order in the bibliography.

- Please ensure that the Article File you upload is a Word file.

Confirmed.

- Please include an Abstract Figure file and an Abstract Figure legend. An appropriate figure legend, which should not exceed 150 words in length, should be included in the main manuscript file. The Abstract Figure is a piece of artwork designed to give readers an immediate understanding of the research and should summarise the main conclusions. If possible, the image should be easily 'readable' from left to right or top to bottom. It should show the physiological relevance of the manuscript so readers can assess the importance and content of its findings. Abstract Figures should not merely recapitulate other figures in the manuscript. Please try to keep the diagram as simple as possible and without superfluous information that may distract from the main conclusion(s). Abstract Figures must be provided by authors no later than the revised manuscript stage and should be uploaded as a separate file during online submission labelled as File Type 'Abstract Figure'. Please also ensure that you include the figure legend in the main article file. All Abstract Figures should be created using BioRender. Authors should use The Journal's premium BioRender account to export high-resolution images. Details on how to use and access the premium account are included as part of this email.

We have included an Abstract figure and legend in the revised submission.

EDITOR COMMENTS

Reviewing Editor:

Three expert referees in the field have reviewed the work. Both referees agreed that a revised manuscript has the potential to significantly impact the field. They each identified concerns that, if addressed, would improve the overall quality and clarity of the work. In addition, the ethics editor identified several points that need to be identified to conform with journal policy.

Thank you for the very constructive reviews.

Please also see 'Required Items' above.

REFEREE COMMENTS

Referee #1:

Thank you for submitting your manuscript to The Journal of Physiology. Some additional details pertaining to animal ethics and welfare are required.

We are grateful to the Referee for reviewing our work.

1. You must start the methods section with the subheading "Ethical approval". Include the institutional approval code/number for the study.

We have included a section **Ethical approval** at the start of the Materials and Methods section including the Home Office project licence number.

2. Please add text to confirm that the study conformed to the ethical principles of The Journal of Physiology.

We confirm that the study conformed to the ethical principles of The Journal of Physiology and have added the text **The study conformed to the ethical principles of The Journal of Physiology.**

3. Provide details on the source of the animals. Briefly describe husbandry arrangements and confirm in the text that animals had free access to food and water.

This is now included: **Adult male Wistar rats (275–325 g) were purchased from Charles River Laboratories (UK). Animals were maintained under controlled temperature (22 ± 2 °C) and 12h light-dark cycle with access to standard chow diet and water *ad libitum*.**

4. Please state killed (or euthanised) by intraperitoneal injection of...

We have added the words **terminally anaesthetised** to our previous text to read: **Animals were terminally anaesthetised by intraperitoneal injection of sodium pentobarbital (200 mg kg⁻¹) and sodium heparin (200 IU kg⁻¹).**

State how death was confirmed.

We have added: **Surgical level of anaesthesia was confirmed by absence of corneal and pedal reflexes and hearts were then excised and immediately arrested...** However, it should be noted that despite a terminal level of anaesthesia, animals are alive at the point of heart excision and therefore death is not confirmed prior to removing the heart using the Langendorff technique.

5. Please note that there are growing concerns with the use of pentobarbitone even for single terminal use as it is an irritant. I note that its use was authorised in the study by the relevant bodies, and therefore is fully acceptable, but The Journal is currently encouraging authors to consider more refined approaches for euthanasia for future work, unless there are strong grounds for justification of use and/or steps to refine its use, such as attendant use of lidocaine.

We are grateful for the reviewer bringing this to our attention. We will consider these possible alternatives in our future work and raise the concerns to our local preclinical operation manager.

Referee #2:

1. Please clarify novelty in plain terms. Explicitly define in the Abstract or Introduction that "this is the first method to independently and quantitatively resolve sarcolemmal and mitochondrial membrane potentials in vivo using dynamic radiotracer modelling."

We are grateful to the Referee for reviewing our work.

We have added the requested statement at the beginning of the last paragraph of the introduction: **To our knowledge, this is the first time that dynamic pharmacokinetic modelling of lipophilic cationic radiotracers has been used to independently and quantitatively resolve sarcolemmal and mitochondrial membrane potentials in vivo.**

2. Provide a succinct paragraph discussing potential deviations from the ideal Nernst equilibrium (e.g., non-homogeneous mitochondrial populations, active transporters, or changes in perfusion) and their impact on $\Delta\Psi_m$ estimation.

We assume that the reviewer refers to non-Nernst behaviour of the ^{99m}Tc SPECT tracers rather than the membranes themselves. The Nernst potential is only valid for individual ions and hence the membrane potential is a composite of the different ion concentrations and permeabilities given by the GHK equation. The membranes themselves therefore never follow ideal Nernst behaviour. For the tracer we assume that its distribution is governed by the voltage and therefore the Nernst equation is an inherent assumption of our model. There are two major potential causes for a deviation from ideal-Nernst behaviour, non-specific binding meaning that chemical activities are not the same as concentrations, and active transport processes that oppose the electrochemical gradient, for example P-gp expression. We have commented on both possibilities,

The Nernst equation should be written in terms of chemical activities not concentrations and therefore deviations from ideal behaviour are possible, as pointed out by the Referee. We had previously discussed this point on p6:

Note that the Nernst equation is expressed in terms of the reaction quotient of chemical activities a_{out}/a_{in} which accounts for non-ideal behaviour. However, noting that the concentrations of radioactive species are typically in the nM range, see Methods, then the chemical activities are equal to chemical concentrations in very dilute solutions, and the reaction quotient is equal to the equilibrium constant K .

Which we have expanded to include further caveats to ideal behaviour:

Non-specific binding of the tracer inside the cell could also lead to a deviation from ideal behaviour.

We further point out in the results that:

For the highest concentration of 600 nM CCCP there was almost complete washout of both tracers.

And in the Discussion:

Treatment of hearts with CCCP led to significantly increased washout rates of both tracers. Almost complete washout of the tracers at the highest concentration of 600 nM CCCP suggests that there is minimal nonspecific binding that would be a confounding factor in the estimation of membrane potentials.

A further possible confounding factor in this regard is the expression of active transporters, for example P-glycoproteins which are ATP dependent efflux transporters, may also lead to non-ideal behaviour. We have added to the Theory section

P-glycoprotein (P-gp) expression can also lead to efflux of the tracer in some disease models which may also lead to non-ideal behaviour, however basal expression of P-gp in the myocardium is very low (Couture et al., 2006).

And to the Study limitations and future direction section:

^{99m}Tc is a known substrate for p-glycoprotein (P-gp), an ATP dependent efflux pump that is important for the biodistribution of pharmaceutical agents, assessment of liver and kidney toxicity and contributes to drug resistance, for example in cancer. P-gp is also a major component of the blood brain barrier and therefore it is interesting to note that brain uptake of [^{99m}Tc]sestamibi seen in Figure 5 is low. High levels of P-gp expression would pump [^{99m}Tc]sestamibi out of the cell leading to increased washout rates. A major application of [^{99m}Tc]sestamibi in the clinic is for this very purpose. Therefore, high expression levels of P-gp in certain tissues such as liver, kidney or in some cancers may confound the measurement of membrane potentials in certain tissues and organs. Expression levels are much lower in the heart. The data presented here strongly supports that the mechanism for retention and pharmacokinetics of [^{99m}Tc]sestamibi and [^{99m}Tc]tetrofosmin in the heart are determined by the membrane voltages. Further experiments will be required to determine the importance of P-gp expression on the pharmacokinetics.

3. The Discussion should include 2-3 sentences on how this framework could be adapted for other tissues (brain, liver) or diseases (diabetes, neurodegeneration), thus underlining its conceptual breadth.

We have not performed the experiments to test the approach in different disease areas. However, please see additional text in answer to comment 2 regarding other tissues and organs in the body. For example, we note that brain uptake is rather low in our images and therefore is likely not be amenable to analysis using our method. We hope this is sufficient at this stage without further speculation.

4. While figures are clear, including a schematic summarizing how E_m and $\Delta\Psi_m$ are mathematically derived from tracer kinetics (Equation links \rightarrow model \rightarrow MCMC output) would improve readability for physiologists less familiar with radiokinetic modelling.

We have added a schematic flow diagram of the fitting procedure in the Supplementary Information file. This describes the different steps in the MATLAB script which has also been updated in the Supplementary Information file. We have also included some further detail on the modelling in the Materials and Methods section of the manuscript.

5. Throughout the manuscript, ensure consistent notation and sign conventions for potentials (negative vs positive voltages). Please clarify in the Methods that E_m and $\Delta\Psi_m$ are defined as negative inside the membrane for interpretive consistency.

Confirmed. We have clarified this point bottom of p6: Where E_m and $\Delta\Psi_m$ are the sarcolemmal and mitochondrial membrane potentials, respectively, which are defined as negative voltages (inside the membrane) in the above equations.

6. The reported SDs (± 1 mV for E_m ex vivo) seem surprisingly low given experimental and modelling variability. Please discuss the possible underestimation of uncertainty due to correlated parameters or limited sample size.

We thank the reviewer for highlighting this point. The reviewer is correct that this SD is rather low compared to the other measurements we reported and may not reflect technical reproducibility of the fitting. We note that we had previously reported errors as biological reproducibility and not previously taken account of the technical variability in the fitting which should have also been accounted for in the error estimation. In revising this point, we have gone back to the raw data and have made some improvements to the MCMC fitting procedure to give a more robust estimation of the errors. Consequently, we have reanalysed ALL the data and updated all the figures. The parameter estimates are largely unchanged; however, this has altered the error estimates which are now more consistent, the errors are larger than previously quoted as they now reflect the difficulties in performing such multiparametric fits. Thus, for a set of data with a fitting variance of S_{fit}^2 and a biological variance of S_{exp}^2 the compounded standard deviation should be quoted as $SD = \sqrt{S_{fit}^2 + S_{exp}^2}$. We have corrected this throughout. We are grateful to the reviewer for pointing out this oversight.

7. Strengthen the last paragraph of the Discussion by explicitly connecting this technique to clinical endpoints, such as assessing myocardial viability, mitochondrial dysfunction, or early cardiotoxicity in oncology patients.

We have added a paragraph at the end of the Discussion section: Methods such as the one proposed here, and others in the literature, have great potential to inform clinical endpoints. Assessing myocardial viability has been an important application in clinical cardiovascular MRI. More generally mitochondrial dysfunction is a key component of cardiovascular pathologies as well as many other disease aetiologies. Evidence also points to the emerging role that mitochondrial dysfunction plays in cardiotoxicity in oncology patients during or after treatment with chemotherapies.

8. Abstract: Add a one-sentence concluding statement highlighting clinical and physiological implications.

See previous response to point 7. We believe this also responds to this point.

9. Introduction: Replace phrases like "we developed a modelling framework" with "we established a quantitative imaging paradigm," which conveys broader conceptual weight.

We have added: We established a quantitative kinetic modelling framework...

10. Discussion: Combine the "Study Limitations" section with a forward-looking "Future Directions" paragraph for smoother flow.

We have renamed the section: **Study Limitations and Future Directions**

11. References: Include recent work on radiolabeled mitochondrial probes (e.g., 2024-2025 total-body PET studies) to emphasize current relevance.

We cited some of the key papers on total-body PET. However, we have added a further recent 2025 citation to emphasize current relevance (Cook et al., 2025).

12. Supplementary Materials: Consider including convergence plots or trace diagnostics for MCMC to reassure readers of parameter stability.

We agree that it would be useful to include the MCMC plots for the experimental data to reassure readers of parameter stability. We have substituted the data in Fig 2 for an experimental control perfused heart dataset rather than the previous simulated data. We have added the MCMC trajectories and corresponding histograms to figures 2, 3, 4 and 5 to better appreciate parameter stability. We have also added the [^{99m}Tc]tetrafosmin data as a supplemental Fig S1 which also includes the diagnostic MCMC trajectories as we had previously not presented this data. We hope this addresses the referee's comment.

Referee #3:

The authors propose and test a modelling approach to estimate sarcolemmal (E_m) and mitochondrial ($\Delta\Psi_m$) membrane potentials in rat hearts using planar scintigraphy with [^{99m}Tc]sestamibi and [^{99m}Tc]tetrofosmin. By combining first-pass pharmacokinetics, the Nernst equation, and Markov Chain Monte Carlo fitting, they derive time-averaged potentials ex vivo in Langendorff-perfused hearts and in vivo, and compare these values with published data. The approach is conceptually elegant and in particular from the modelling perspective methodologically innovative. The following comments aim to help clarify specific methodological aspects and strengthen the interpretation and translational scope.

We are pleased that the reviewer found our work to be innovative and elegant. Thank you.

RC#1: The definition of the "first-pass" period used for kinetic modelling remains ambiguous. In Fig. 4, model fits extend to ~1000 s, suggesting that data beyond the first-pass transit may have been included in parameter estimation. Because tracer retention after initial uptake is influenced by mitochondrial binding and slow washout rather than instantaneous membrane potentials, the authors should clarify a. the exact temporal range of data used for fitting (preferably also indicated in the figure), b. whether later time points were weighted or excluded, and c. how this choice affects the interpretation of E_m and $\Delta\Psi_m$ as first-pass rather than steady-state parameters.

The perfused heart is inherently a first pass technique in the sense that the radiotracer goes through the heart only once and is not recirculated. It is in this context that we used the term. The referee is correct that the data was acquired beyond what would normally be classed as first pass. Regarding non-specific binding, also noted in our response to referee 2, the experiment with 600 nM CCCP leads to almost complete washout of the tracer. Therefore, it is in fact the voltages that are leading to the unequal distributions in the cytosolic and mitochondrial compartments (governed by the Nernst equation) and not nonspecific binding as the referee suggests. We used all the data for the fitting and did not weight or exclude later timepoints. We have clarified this point in the Methods section: **All data were included in the fitting without any weighting factors at different timepoints; however, this could also be included in future iterations of the model.** Regarding the final point, parameterizing the model according to rate constants makes no assumption about whether the kinetics are in the first pass regime or in the steady state. The model still reaches the steady state at later time points. Therefore, the entire kinetic timecourses are governed by E_m and $\Delta\Psi_m$.

To avoid ambiguity in the terminology we have removed the term 'first pass' since we analysed all the data with equal weighting.

RC#2: The authors have appropriately validated their kinetic model using simulated data and examined parameter correlations qualitatively. To strengthen confidence in the robustness of the voltage estimates, it would be valuable to quantify the sensitivity of E_m and $\Delta\Psi_m$ to variations in the fitted transport rate constants (k_1 , k_2). Given that these parameters are influenced by myocardial perfusion and tracer diffusion characteristics, the authors might compare their fitted values with independent perfusion or extraction data for [^{99m}Tc]sestamibi and [^{99m}Tc]tetrofosmin, or report sensitivity coefficients or correlations derived from the Jacobian. This would clarify the stability of the potential estimates against physiologically plausible variations in tracer kinetics.

We thank the reviewer for this comment. We do not have independent perfusion or extraction data for [^{99m}Tc]sestamibi and [^{99m}Tc]tetrofosmin. However, the Referee is completely right this is likely the most challenging limitation of our method, the dependence of the parameter k_1 on flow *and* transport. We have explored this point through additional simulations and fitting in Fig S2 to investigate the influence of flow rate constant on the derived membrane potentials. Indeed, as pointed out by the referee, when flow is slower than the uptake rate constant then the estimated voltages are subject to far greater uncertainty and are biased with respect to the known values in the simulation. However, we note that under conditions that k_f is greater than k_1 then the MCMC procedure still does a good job estimating the true parameters. It is also of interest to note in these simulations that uptake becomes independent of flow at higher flow rates, therefore in keeping with the literature on the mechanism of uptake for [^{99m}Tc]sestamibi.

RC#3: While the study convincingly demonstrates the feasibility of estimating sarcolemmal and mitochondrial membrane potentials in vivo using radiotracer kinetics, the manuscript would benefit from a clearer discussion of the temporal and spatial resolution limitations inherent to the approach. In contrast to optical techniques, which in experimental settings allow continuous and subcellular monitoring of membrane potential dynamics, the presented method provides single, first-pass "snapshot" measurements with millimetre-scale spatial resolution due to planar scintigraphy. Explicitly quantifying these constraints and discussing their implications for detecting rapid or regional potential changes would strengthen the reader's understanding of the technique's current scope and translational potential.

We agree that there are many challenges to the technique, including temporal resolution and spatial resolution. The fastest temporal resolution that we achieved in vivo with planar scintigraphy was 1s for the early timepoints, enabling us to sample the input function with sufficient resolution to define the peak activity. However, this required us to do the experiments using planar scintigraphy that has no depth resolution. This will limit the ability to assess regional changes of membrane potential using this technique. We have added a statement in the Limitation section: **Planar scintigraphy has a further limitation in that it offers no depth resolution. It would therefore be challenging to assess regional changes in membrane potential using this technique.** It is for this reason that we had already proposed the solution would be dynamic PET imaging, however this requires a great deal more work to develop lipophilic positron emitting radiotracers that have similar characteristics. Without having done the experiments we prefer not to speculate further at the current time.

Dear Dr Eykyn,

Re: JP-RP-2025-290295R1 "Sarcolemmal and mitochondrial membrane potentials measured *ex vivo* and *in vivo* in the heart by pharmacokinetic modelling of [^{99m}Tc]sestamibi" by Edward C.T. Waters, Friedrich Baark, Matthew R. Orton, Michael J Shattock, Richard Southworth, and Thomas R. Eykyn

Thank you for submitting your manuscript to The Journal of Physiology. It has been assessed by a Reviewing Editor and by 2 expert referees and we are pleased to tell you that it is acceptable for publication following satisfactory revision.

REVISION CHECKLIST:

Please upload two versions of your manuscript text: one with all relevant changes highlighted and one clean version with no changes tracked. The manuscript file should include all tables and figure legends, but each figure/graph should be uploaded as separate, high-resolution files. The journal is now integrated with Wiley's Image Checking service. For further details, see: <https://www.wiley.com/en-us/network/publishing/research-publishing/trending-stories/upholding-image-integrity-wileys->

image-screening-service

We look forward to receiving your revised submission.

Yours sincerely,

Natalia Trayanova
Senior Editor
The Journal of Physiology

REQUIRED ITEMS

- Please include an Abstract Figure legend. An appropriate figure legend, which should not exceed 150 words in length, should be included in the main manuscript file. The Abstract Figure is a piece of artwork designed to give readers an immediate understanding of the research and should summarise the main conclusions. If possible, the image should be easily 'readable' from left to right or top to bottom. It should show the physiological relevance of the manuscript so readers can assess the importance and content of its findings. Abstract Figures should not merely recapitulate other figures in the manuscript. Please try to keep the diagram as simple as possible and without superfluous information that may distract from the main conclusion(s). Abstract Figures must be provided by authors no later than the revised manuscript stage and should be uploaded as a separate file during online submission labelled as File Type 'Abstract Figure'. Please also ensure that you include the figure legend in the main article file. All Abstract Figures should be created using BioRender. Authors should use The Journal's premium BioRender account to export high-resolution images. Details on how to use and access the premium account are included as part of this email.
- You must start the Methods section with a paragraph headed Ethical approval (https://jp.msubmit.net/cgi-bin/main.plex?form_type=display_requirements#methods).

Research must comply with The Journal's policies regarding animal experiments (<https://physoc.onlinelibrary.wiley.com/hub/animal-experiments>) and adherence to these policies must be stated in the manuscript.

Authors should confirm in their Methods section that their experiments were carried out according to the guidelines laid down by their institution's animal welfare committee, including an ethics approval reference number. The Methods section must contain a statement about access to food, water and housing, details of the anaesthetic regime: anaesthetic used, dose and route of administration, and method of killing the experimental animals.

EDITOR COMMENTS

Reviewing Editor:

Methods Details:

Please clarify the terminal procedure for in vivo planar scintigraphy of animals.

Comments to the Author:

Both referees agree that the authors have satisfactorily addressed many of their original concerns. This manuscript is expected to have a significant impact on the field. Referee 1 identified additional opportunities to improve the manuscript. Also, the authors should clearly state the terminal procedures for the in vivo planar scintigraphy.

REFEREE COMMENTS

Referee #1:

Thank you for the substantial revision and the detailed, point-by-point response to the reviewer. The manuscript is clearer

and more convincing following the global reanalysis and the rewritten fitting code, and the key requests from the reviewer appear to have been addressed: the novelty is now stated explicitly in plain terms, the kinetic-to-voltage workflow is easier to follow (including a schematic/flow diagram in the Supplementary Information), notation and sign conventions for E_m and $\Delta\Psi_m$ are clarified, and the previously implausibly small uncertainties are now treated more appropriately by accounting for technical fitting variance in addition to biological repeatability; the added MCMC trace/histogram diagnostics and the strengthened Discussion linking the framework to clinically relevant endpoints (e.g., viability, mitochondrial dysfunction, and cardiotoxicity) further improve reader confidence and impact. One area that would still benefit from a small but targeted refinement is the reviewer's request for a succinct, consolidated discussion of departures from ideal Nernst behaviour and related modelling assumptions: you now mention activities vs concentrations, non-specific binding, active efflux (e.g., P-gp), and flow/perfusion dependence, but these points are spread across sections; please consider combining them into a single short paragraph that explicitly states that the approach yields an effective, volume-averaged E_m and $\Delta\Psi_m$, and that heterogeneity in mitochondrial populations, regional perfusion, or transporter expression could bias estimates or broaden posteriors (and briefly indicate how such confounds might be detected or mitigated in future work). Similarly, the request to underline conceptual breadth beyond the heart could be satisfied without over-speculation by adding 2-3 sentences noting that the framework is, in principle, transferable to other organs and disease contexts provided tracer uptake is adequate and organ-specific confounds (BBB/low uptake, higher P-gp expression, different extracellular volumes, and the need for 3D dynamic PET/SPECT for regional mapping) are accounted for. With these minor clarifications, the revision should satisfactorily close the reviewer's remaining concerns.

Referee #2:

The authors have addressed my comments and added new relevant data. The low flow simulations and perspective on spatial resolution allow to better understand strengths and limitations of this interesting method.

No further comments.

END OF COMMENTS

Dear Editor and Referees,

We are very grateful for the additional reviews of our work. We are pleased that our revision has satisfied most of the criticisms raised. Please find below response to the final points for clarification by Referee #1. Our responses are in green text and additions to the manuscript text in red.

EDITOR COMMENTS

Reviewing Editor:

Methods Details:

Please clarify the terminal procedure for in vivo planar scintigraphy of animals.

Comments to the Author:

Both referees agree that the authors have satisfactorily addressed many of their original concerns. This manuscript is expected to have a significant impact on the field. Referee 1 identified additional opportunities to improve the manuscript. Also, the authors should clearly state the terminal procedures for the in vivo planar scintigraphy.

Thank you. We have added this to the Materials and Methods section.

At the end of scanning animals were culled by overdose of anaesthetic.

REFEREE COMMENTS

Referee #1:

Thank you for the substantial revision and the detailed, point-by-point response to the reviewer. The manuscript is clearer and more convincing following the global reanalysis and the rewritten fitting code, and the key requests from the reviewer appear to have been addressed: the novelty is now stated explicitly in plain terms, the kinetic-to-voltage workflow is easier to follow (including a schematic/flow diagram in the Supplementary Information), notation and sign conventions for E_m and $\Delta\Psi_m$ are clarified, and the previously implausibly small uncertainties are now treated more appropriately by accounting for technical fitting variance in addition to biological repeatability; the added MCMC trace/histogram diagnostics and the strengthened Discussion linking the framework to clinically relevant endpoints (e.g., viability, mitochondrial dysfunction, and cardiotoxicity) further improve reader confidence and impact. One area that would still benefit from a small but targeted refinement is the reviewer's request for a succinct, consolidated discussion of departures from ideal Nernst behaviour and related modelling assumptions: you now mention activities vs concentrations, non-specific binding, active efflux (e.g., P-gp), and flow/perfusion dependence, but these points are spread across sections; please consider combining them into a single short paragraph that explicitly states that the approach yields an effective, volume-averaged E_m and $\Delta\Psi_m$, and that heterogeneity in mitochondrial populations, regional perfusion, or transporter expression could bias estimates or broaden posteriors (and briefly indicate how such confounds might be

detected or mitigated in future work). Similarly, the request to underline conceptual breadth beyond the heart could be satisfied without over-speculation by adding 2-3 sentences noting that the framework is, in principle, transferable to other organs and disease contexts provided tracer uptake is adequate and organ-specific confounds (BBB/low uptake, higher P-gp expression, different extracellular volumes, and the need for 3D dynamic PET/SPECT for regional mapping) are accounted for. With these minor clarifications, the revision should satisfactorily close the reviewer's remaining concerns.

Thank you for the additional clarifications. We have added the following text to the Study Limitations and Future Directions section without adding further speculation at this stage.

While the approach yields an effective, volume and time-averaged E_m and $\Delta\Psi_m$, heterogeneity in mitochondrial populations, regional perfusion, or transporter expression could bias estimates or broaden posteriors. ^{99m}Tc is a known substrate for p-glycoprotein (P-gp), an ATP dependent efflux pump that is important for the biodistribution of pharmaceutical agents, assessment of liver and kidney toxicity and contributes to drug resistance, for example in cancer. P-gp is also a major component of the blood brain barrier and therefore it is interesting to note that brain uptake of [^{99m}Tc]sestamibi seen in Figure 5 is low. High levels of P-gp expression would pump [^{99m}Tc]sestamibi out of the cell leading to increased washout rates. A major application of [^{99m}Tc]sestamibi in the clinic is for this very purpose. Therefore, high expression levels of P-gp in certain tissues such as liver, kidney or in some cancers may confound the measurement of membrane potentials in certain tissues and organs. Expression levels are much lower in the heart. The data presented here strongly supports that the mechanism for retention and pharmacokinetics of [^{99m}Tc]sestamibi and [^{99m}Tc]tetrofosmin in the heart are determined by the membrane voltages. In principle, the framework is transferable to other organs and disease contexts provided tracer perfusion is adequate and organ-specific confounds are accounted for such as BBB/low uptake, higher P-gp expression, different extracellular volumes, and the need for 3D dynamic PET/SPECT for regional mapping. Further experiments will be required to determine the importance of P-gp expression on the pharmacokinetics. Possible deviations from ideal Nernst behaviour are also possible where molar activities do not equate to concentration, particularly non-specific binding which would perturb the equilibrium distribution across the membrane.

Referee #2:

The authors have addressed my comments and added new relevant data. The low flow simulations and perspective on spatial resolution allow to better understand strengths and limitations of this interesting method.

Thank you.

No further comments.

Dear Dr Eykyn,

Re: JP-RP-2026-290295R2 "Sarcolemmal and mitochondrial membrane potentials measured *ex vivo* and *in vivo* in the heart by pharmacokinetic modelling of [^{99m}Tc]sestamibi" by Edward C.T. Waters, Friedrich Baark, Matthew R. Orton, Michael J Shattock, Richard Southworth, and Thomas R. Eykyn

We are pleased to tell you that your paper has been accepted for publication in The Journal of Physiology.

Yours sincerely,

Natalia Trayanova
Senior Editor
The Journal of Physiology

IMPORTANT POINTS TO NOTE FOLLOWING ACCEPTANCE OF YOUR PAPER:

- **IMPORTANT NOTICE ABOUT OPEN ACCESS:** To assist authors whose funding agencies mandate immediate public access to published research findings, The Journal of Physiology allows authors to pay an Open Access (OA) fee to have their papers made freely available immediately on publication.

- You can help your research get the attention it deserves! Check out Wiley's free Promotion Guide for best-practice recommendations for promoting your work at: www.wileyauthors.com/eoo/guide. You can learn more about Wiley Editing Services which offers professional video, design, and writing services to create shareable video abstracts, infographics, conference posters, lay summaries, and research news stories for your research at: www.wileyauthors.com/eoo/promotion.

- If you would like to receive our 'Research Roundup', a monthly newsletter highlighting the cutting-edge research published in The Physiological Society's family of journals (The Journal of Physiology, Experimental Physiology, Physiological Reports, The Journal of Nutritional Physiology and The Journal of Precision Medicine: Health and Disease), please click this link, fill in your name and email address and select 'Research Roundup':
<https://www.physoc.org/journals-and-media/membernews>

EDITOR COMMENTS

Reviewing Editor:

Comments to the Author:

The authors have addressed the referee's concerns. The manuscript is expected to have a significant impact on the field.